# Follow Hamiltonian Leader: An Efficient Energy-Guided Sampling Method

## Abstract

Our research underscores the value of leveraging zeroth-order information for addressing sampling challenges, particularly when first-order data is unreliable or unavailable. In light of this, we have developed a novel parallel sampling method that incorporates a leader-guiding mechanism. This mechanism forges connections between multiple sampling instances via a selected leader, enhancing both the efficiency and effectiveness of the entire sampling process. Our experimental results demonstrate that our method markedly expedites the exploration of the target distribution and produces superior quality outcomes compared to traditional sampling techniques. Furthermore, our method also shows greater resilience against the detrimental impacts of corrupted gradients as intended.

## 1 Introduction

Score-based generative models [35, 26, 36, 20] introduce a novel approach to generative modeling that revolves around the estimation and sampling of the Stein score [26, 36]. The score represents the gradient of the log-density function $\nabla_x \log \pi(x)$ evaluated at the input data point $x$. This type of approach usually relies on effectively training a deep neural network to accurately estimate the score. The estimated score is then utilized to navigate the sampling process, ultimately resulting in the production of high-quality data samples that closely match the areas of high density in the original distribution.

In our research, we investigate the sampling of a probability distribution given by $\pi(x) \propto e^{-U(x)}$, where $U(x)$ is the energy function. In the context of energy-based score-matching generative models, the objective often involves sampling the modes in areas of high probability density. An approach as suggested in [36, 20], is to smooth the original distribution by convolving $\pi(x)$ with an isotropic Gaussian distribution of variance $\sigma^2$, yielding $\pi_\sigma(x) = \int \pi(x')\mathcal{N}(x; x', \sigma^2 I) \, dx'$. By gradually decreasing the variance $\sigma$, $\pi_\sigma(x)$ recovers the original distribution $\pi(x)$.

Typically, the sampling of score-based approaches are integrated with numerical SDE solvers [38], for example, the Euler-Maruyama solver, as well as Monte Carlo Markov Chain (MCMC) techniques like Langevin Dynamics [30]. Furthermore, there is a notable similarity between score-based sampling algorithms and first-order optimization algorithms. Efforts have been made to merge these two methodologies, particularly from a perspective of sampling [42, 10, 28, 9, 44]. All these methods primarily concentrates on first-order information $\nabla_x U(x)$ to improve performance, while typically treating the zeroth-order information $U(x)$ merely as a basis for rejecting samples [18, 32, 29].

We argue that incorporating zeroth-order information can significantly enhance the algorithm's overall effectiveness, particularly in instances where the first-order information is compromised. To address this, we draw inspiration from parallel tempering [39], a simulation method commonly used to identify the lowest energy state in systems of interacting particles. The fundamental principle of

parallel tempering involves operating multiple sampling replicas simultaneously, each at a different temperature level. These temperatures typically range from low, where the system is prone to being trapped in local minima, to high, which facilitates the system's ability to surmount energy barriers and more thoroughly explore the energy landscape.

Drawing inspiration from this concept, we extend the Hamiltonian Monte Carlo (HMC) framework [29] and introduce a novel algorithm that concurrently runs multiple replicas, sampling at both high and low Hamiltonian energy levels. Moreover, this methodology combines both zeroth and first order information from various chains, hence enhancing the effectiveness of sampling approaches. The experimental findings demonstrate the efficacy of our approach in scenarios where relying solely on first-order knowledge is insufficient. These findings illustrate the capacity of incorporating zeroth-order information to greatly enhance the efficiency and accuracy of sampling operations in energy-based score-matching algorithms.

## 2 Background

### 2.1 Hamiltonian Monte Carlo

The primary purpose of MCMC is to construct a Markov chain that matches its equilibrium distribution to the target distribution. One of the most popular MCMC methods is Langevin Monte Carlo [17, 32], which proposes samples in a Metropolis-Hastings [18] framework for more efficient state space exploration. Another advanced method is HMC [29, 11, 2], which incorporates an auxiliary variable $p$ and employs Hamiltonian dynamics to facilitate the sampling process. The Hamiltonian function is structured as a composite of potential energy $U(x)$ and kinetic energy $K(p)$, defined as follows:

$$H(x, p) = U(x) + K(p), \tag{1}$$

where $x$ represents the position of a particle and $p$ denotes its momentum. Kinetic energy $K(p)$ is commonly formulated as $K(p) = \frac{1}{2}p^T M^{-1} p$, where $M$ corresponds to the mass matrix. For simplicity, we assume in this paper that the mass matrix $M$ is equal to the identity matrix $I$. The joint distribution of position and momentum conforms to the canonical distribution:

$$\pi(x, p) = e^{-H(x,p)}/Z, \tag{2}$$

where $Z = \iint e^{-H(x,p)} \, dx dp$. Samples from $\pi(x)$ can then be obtained by marginalizing $p$ from $\pi(x, p)$, which further requires $\int_p \pi(x, p) \, dp = $ constant. In the HMC algorithm, proposals are generated by simulating Hamiltonian dynamics and then subjected to a Metropolis criterion to determine their acceptance or rejection. A commonly employed numerical method for solving these equations is the *Leapfrog* integrator [3].

Recent progress in HMC techniques has focused on increasing their adaptability and applicability in a variety of contexts. Such developments include the NUTS sampler [21], which features an automatic mechanism for adjusting the number of simulation steps. The Riemann manifold HMC [15] leverages Riemannian geometry to modify the mass matrix $M$, making use of curvature information to improve sampling efficiency. Additionally, Stochastic Gradient Hamiltonian Monte Carlo [11, 27] investigates a stochastic gradient approach within the HMC framework. Our contribution is distinct from these methods and can be easily integrated with them.

### 2.2 Energy-based score-matching model

Probabilistic models often require normalization, which can become infeasible when dealing with high-dimensional data [25, 13]. Since the exact probabilities of less probable alternatives become less crucial as long as they remain relatively lower, rather than solely predicting the most probable outcome, models can be structured to interpret relationships between variables via an energy function. Within the context of generative models, these energy-based models (EBMs) are devised to assign higher energy values to regions of lower probability and lower energy values to regions of higher probability.

Score matching [22, 36] is a method used in statistical modeling and machine learning to estimate a probability distribution or a probability density function from observed data. It is particularly useful when direct estimation of the probability distribution is challenging, especially in high-dimensional spaces. In score matching, the goal is to find an approximation to the probability density function (PDF) of a dataset by estimating the score function, also known as the gradient of the log-density.

The score function represents the derivative of the log PDF with respect to the data. By matching the estimated score function to the observed data, one can indirectly estimate the underlying probability distribution.

A relationship between EBMs and score matching can be established by training EBMs through denoising score matching [37]. The training objective is described below:

$$\mathbb{E}_{\pi(x)\mathcal{N}(\epsilon;0,I)}\left[\left\|\frac{\epsilon}{\sigma}-\nabla_x U_\theta\big(x+\sigma\epsilon\big)\right\|_2^2\right]. \tag{3}$$

$U_\theta$ is typically represented as a neural network, with $\theta$ denoting its parameters. Minimizing this objective ensures that $\nabla_x U_\theta(x) = -\nabla_x \log \pi_\sigma(x)$ and thus $e^{-U_\theta(x)}$ shall be proportional to $\pi_\sigma(x)$.

## 3 Motivation

In our work, we assume to have access to both the gradient information $\nabla_x U(x)$ as well as the energy information $U(x)$. In certain scenarios, gradients may yield information that is either of limited or potentially detrimental. Our research examines situations where gradients are compromised, highlighting the importance of zeroth-order information, often associated with energy-based sampling.

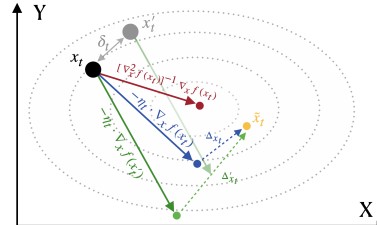

Figure 1: A good *anchor point* could help improve convergence even if the gradient is *unexpectedly disturbed* from *original gradient* to the *disturbed gradient*, getting closer to the optimal point.

We concentrate on showcasing the strengths of our method in three types of challenging but common scenarios, summarized as *instability*, *metastability* and *pseudo-stability*. Instability refers to a state in which a system lacks equilibrium or steadiness, often leading to unpredictable or erratic behavior. Metastability describes a condition where a system appears stable over a short period but is not in its most stable state, and it can transition to a more stable state under certain conditions. Pseudo-stability, on the other hand, denotes a situation where a system seems stable but is actually in an incorrect, suboptimal, or misleadingly stable state.

**Instability.** In high-dimensional spaces, sampling algorithms may struggle to converge in the presence of a complex probability distribution. This instability can arise in situations where the local Hessian matrix is ill-conditioned or spectrum of the local Hessian matrix is exceptionally large. Such conditions often lead to inaccuracies or instabilities in numerical calculations, potentially causing the convergence process to fail. The samples generated could substantially diverge from the true mode, resulting in subpar sample quality. However, employing an anchor point can enhance the stability of convergence, as demonstrated in Figure 1.

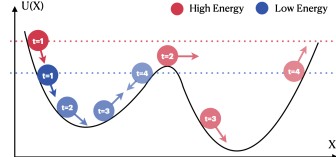

Figure 2: To enhance exploratory capabilities, it's important to encourage particle to explore the landscape.

**Metastability.** Particles are prone to getting stuck in local minima when the gradients are not informative. For example, on the saddle point or a pleaute loss landscape. As a result, simulations frequently end up in a state of intermediate energy, which is different from the system's lowest energy state. This scenario is illustrated in Figure 2.

**Pseudo-Stability.** Certain situations may present a divergence between the gradient information and the ground truth. This divergence can hinder algorithms from accurately converging to the appropriate modes. In these instances, it becomes essential to incorporate energy information to rectify inaccuracies that arise from solely depending on gradients. An example of misleading gradients could be observed in Figure 3.

## 4 Algorithm

Many sampling methods typically rely on independent Markov chains, which can lead to the issues mentioned in Section 3. Taking inspiration from [39], our approach involves the utilization of multiple replicas. This approach

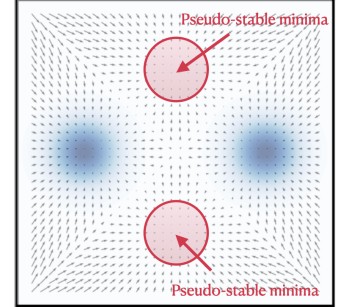

Figure 3: There is a potential for particles to unintentionally follow the gradient flow towards these regions of high energy. A more comprehensive description could be found at Section 5.1.3.

---

**Algorithm 1** Elastic Leapfrog (eLeapfrog)

---

**Input:** A collection of positions $\{x^i\}_{i=1}^n \in \mathbb{R}^{n \times d}$, a collection of momenta $\{p^i\}_{i=1}^n \in \mathbb{R}^{n \times d}$, learning rate $\eta > 0$, pulling strength $\lambda \geq 0$, number of Leapfrog steps $L$.

**for** $s = 1, \cdots, L$ **do**
    **for** $i = 1, \cdots, n$ **do**
        Choose the leader $x^l$ and calculate $\rho^i$
        $g^i \leftarrow \nabla_x U(x^i) + \rho^i \cdot (x^i - x^l); p^i \leftarrow p^i - \frac{\eta}{2} \cdot g^i$          ▷ Half step for momentum
        $x^i \leftarrow x^i + \eta \cdot p^i$          ▷ Full step for position
        Choose the leader $x^l$ and calculate $\rho^i$
        $g^i \leftarrow \nabla_x U(x^i) + \rho^i \cdot (x^i - x^l); p^i \leftarrow p^i - \frac{\eta}{2} \cdot g^i$          ▷ Half step for momentum
    **end for**
**end for**
**Output:** $x \in \mathbb{R}^d, p \in \mathbb{R}^d$

---

enables us to implicitly encourage greater exploration among multiple particles while simultaneously preserving the optimal outcomes for exploitation purposes. We will elaborate on how our algorithm can be employed to tackle these challenges.

Firstly, we introduce a modified version of the leapfrog method, called the *Elastic Leapfrog* (eLeapfrog). In this approach, additional elastic forces are applied between each particle and a *leader*, incorporating an extra elastic energy term into the traditional Hamiltonian function. This modification aims to prevent particles from straying significantly from each other, thereby promoting local exploitation. We then divide the particles into groups and designate the particle with the lowest energy as the leader. Moreover, when combined with the eLeapfrog method, this approach encourages other particles to explore around the leader, efficiently addressing the problem of **instability**.

Due to the properties of HMC, introducing such an extra elastic energy term when pulling the particles towards the leader implicitly incorporates this energy into the momentum, thereby increasing the search ability of each particle. As a result, non-leading particles gain more energy for exploration, while the leading particle is more likely to concentrate on local exploitation. This approach helps mitigate the issue of **metastability**.

Finally, we integrate these techniques to present our compplete *Follow Hamiltonian Leader* (FHL) algorithm. The FHL algorithm capitalizes on both first-order and zeroth-order information while significantly improving the efficiency of space sampling compared to traditional sequential sampling methods. This enhanced approach fosters convergence towards the lowest energy states and increases the likelihood of escaping states with **pseudo stability**.

## 4.1 Elastic Leapfrog

To improve the efficiency of sampling, we integrate an elastic force component into the conventional leapfrog technique. This enhancement aims to dynamically guide particles towards a leading particle, facilitating their movement and improve their exploration ability. The method could be treated like temporarily storing potential energy within an elastic spring, which is then converted into kinetic energy. By adding extra elastic force, we could define the energy of elastic HMC as:

$$H_e(x, p; \tilde{x}) = U_e(x; \tilde{x}) + K(p) = \underbrace{[U(x) + E(x; \tilde{x})]}_{U_e(x, \tilde{x})} + K(p), \tag{4}$$

where $E(x; \tilde{x})$ is the extra elastic energy imposed by Elastic Leapfrog and is defined as $E(x; \tilde{x}) = \frac{\varrho}{2} \|x - \tilde{x}\|_2^2$. Our approach enables particles to efficiently navigate the sample space, guided by the leader. This local exploration strategy, though similar to concepts in [46, 7, 8, 40], is uniquely tailored for application in the realm of sampling.

## 4.2 Leader Pulling

Next, we introduce our *leader pulling* method. Initially, we represent the $i^{th}$ particle inside a group as $x^i$ and select a leader based on a their energies $U(x^i)$. The motivation is that we encourage each

particle $x^i$ to be guided towards a chosen leader. The leader is chosen as the one of minimum energy and thus its index is $l = \arg\min_i U(x^i)$. The objective function for a group of $n$ particles is:

$$U_e(x^1, \cdots, x^n; x^l) = \sum_{i=1}^{n} U(x^i) + \frac{\rho^i}{2} \cdot \|x^i - x^l\|_2^2, \tag{5}$$

where $\pi^i = \exp\left(-U(x^i)\right) / \sum_j \exp\left(-U(x^j)\right)$ and $\rho^i = \lambda \cdot (\pi^l - \pi^i)/(\pi^l + \pi^i)$. The specifics of the *Elastic Leapfrog* algorithm combined with leader pulling technique are detailed in Algorithm 1.

### 4.3 Follow Hamiltonian Leader

Incorporating zeroth-order information (i.e., function values rather than derivatives) serves two key purposes. Firstly, it provides a search direction that accelerates convergence and helps mitigate issues arising from corrupted first-order information (i.e., gradient inaccuracies), thereby speeding up the optimization process. Second, it helps ensure that we are sampling from the correct underlying distribution by properly accepting or rejecting the proposal.

To ensure that the sampling method maintains detailed balance—a requirement for most sampling algorithms—we evaluate the joint distribution of a group of particles. This evaluation determines whether to accept or reject a proposed move for the whole group, thereby preserving the integrity of the sampling process. This adaptation results in the creation of our algorithm FHL, extensively elucidated in Algorithm 2.

---

**Algorithm 2** Follow Hamiltonian Leader

**Input:** A collection of positions $\{x^i\}_{i=1}^n \in \mathbb{R}^{n \times d}$, learning rate $\eta > 0$, pulling strength $\lambda \geq 0$, number of steps $L$.

**for** $t = 1, 2, \cdots, T$ **do**
    # Run sampling in parallel
    **for** $i = 1, \cdots, n$ **do**
        Randomly sample the momentum $p_{t-1}^i \sim \mathcal{N}(0, I)$
        $x_{\text{prop}}^i, p_{\text{prop}}^i \leftarrow \text{eLeapfrog}\left(x_{t-1}^i, p_{t-1}^i, \eta, \lambda, L\right)$
    **end for**
    Sample a random variable $u \sim \text{Uniform}(0, 1)$
    **if** $u < \prod_{i=1}^n \exp\left(H(x_{\text{prop}}^i, p_{\text{prop}}^i) - H(x_{t-1}^i, p_{t-1}^i)\right)$ **then**
        **for** $i = 1, \cdots, n$ **do** $x_t^i \leftarrow x_{prop}^i$, $p_t^i \leftarrow p_{\text{prop}}^i$ **end for**
    **else**
        **for** $i = 1, \cdots, n$ **do** $x_t^i \leftarrow x_{t-1}^i$, $p_t^i \leftarrow p_{t-1}^i$ **end for**
    **end if**
**end for**
**Output:** $X_T = \{x_T^i\}_{i=1}^n \in \mathbb{R}^{n \times d}$

---

## 5 Experiment

In this section, we showcase the efficacy of incorporating zeroth-order information, specifically energy information, into our proposed method to improve the sampling process. We focus on demonstrating the advantages of our approach in addressing the benefits of our approach in handling three distinct adversarial gradient scenarios, as outlined in Section 3. To evaluate our method on the performance of the concerned questions, we conduct a comparative analysis against the following baseline algorithms:

- **LMC (Langevin Monte Carlo)**: An MCMC method as described in [17] that uses Langevin dynamics to sample from probability distributions. It is also known as the Metropolis-adjusted Langevin algorithm.

- **HMC (Hamiltonian Monte Carlo)**: An MCMC algorithm that employs Hamiltonian dynamics for more efficient traversal of the state space, leading to better exploration and sampling from complex distributions [29, 11, 2].

- **U-LMC (Unadjusted Langevin Dynamics)**: A variation of LMC without the Metropolis correction, referred to [32, 1, 42].

- **U-HMC (Unadjusted Hamiltonian Monte Carlo)**: A form of HMC that excludes the Metropolis correction step, as in [34, 14].

## 5.1 Motivating Examples

We report on results addressing the challenges identified as *instability*, *metastability*, and *pseudo-stability*. Our findings lead us to conclude that the FTH method consistently outperforms other approaches in all scenarios examined. Detailed discussions and further analyses of these findings will be presented in the following subsections.

In our experiment, we simultaneously execute sampling with $N$ particles, each completing a total of $T$ sampling steps. For the FTH method, these particles are divided into $N/n$ groups, with each group containing $n$ particles. Throughout all experiments, we set $n$ to 4. For hyperparameter search, we select step sizes $\eta = \{0.002, 0.0002, 0.005, 0.0005\}$ for all methods and number of leapfrog steps $L = \{4, 8, 16\}$ for HMC-type methods.

### 5.1.1 Instability

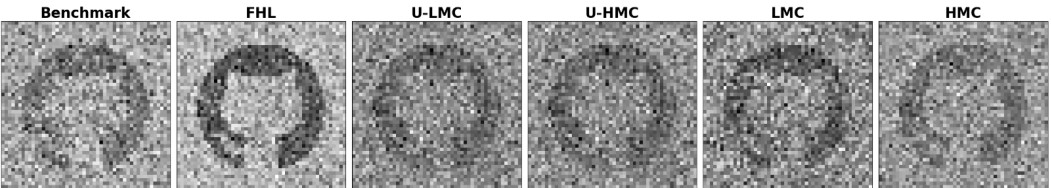

(a) Sampling from the **original distribution** in the form of $e^{-U(x)}$. $T = 1000$ in all experiments.

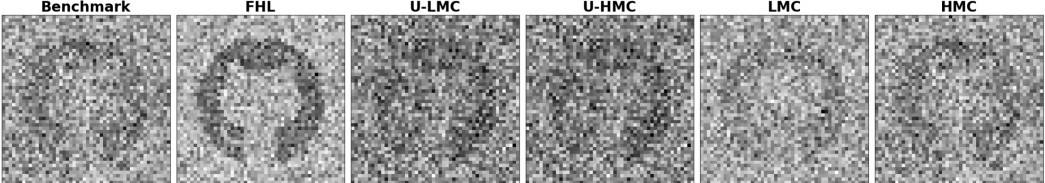

(b) Sampling from the **approximated distribution** in the form of $e^{-U_\theta(x)}$. $T = 2000$ in all experiments.

Figure 4: Sample from a Gaussian distribution $\mathcal{N}(\mu, \Sigma)$ where $\mu \in \mathbb{R}^d$ corresponds to the clean image. For each method, we plot the lowest-energy particle (in terms of $U(x)$ among all particles in $X_T$). The upper-left image represents a direct sample from the distribution $\mathcal{N}(\mu, \Sigma)$; The lower-left image is generated by performing HMC sampling for $T$ steps on the function $U_\theta(x)$, with an initial point set to $x_0 = \mu$.

In our sampling process, we focus on efficiently directing particles to high probability density regions, thereby avoiding unproductive exploration in regions with low probability. When sampling from a single image, our goal becomes attaining the global optima, aligning this objective with those found in optimization tasks.

For our experiment, we chose an image resembling the *GitHub* logo (https://github.com/logos), converted it into a vector format, and use this as the mean of a multivariate Gaussian distribution. The covariance matrix for this distribution, represented by $\Sigma$, is diagonal. The variance for each dimension of the distribution is randomly determined by a uniform distribution within the range of $(0.25, 1.25)$. We carry out two similar but different types of experiments:

In the first experiment, we focus on sampling from the original distribution. This distribution is described mathematically as $e^{-U(x)} \propto \mathcal{N}(\mu, \Sigma)$, with $U(x)$ being the energy function that characterizes the system.

For the energy-based score-matching model, we employ a ResNet [19] architecture with 6 layers of a hidden dimension of 256. The results of the sampling process are detailed in Figure 4, where the main objective is to assess the particles' capacity for effective convergence to the mode of the distribution. In the first scenario, $U(x)$ represents a convex function, whereas in the second scenario, $U_\theta(x)$ is presumed to be non-convex. The findings demonstrate that our approach, FHL, surpasses other baseline methods in both situations.

### 5.1.2 Metastability

Our research explores the concept of metastability, which arises in specific scenarios. Metastability refers to a state of intermediate energy in a dynamic system, differing from its lowest energy state. We examine an extreme scenario where gradients are entirely absent, and sampling methods only get access to the energy information about the distribution.

In Figure 5, it's evident that in this particular situation, we enforce the gradient to be *near zero*, resulting in all sampling methods, except FHL and LMC, behaving almost like random sampling. Nevertheless, owing to the leader pulling strategy, FHL retains its ability to locate the mode much faster.

### 5.1.3 Pseudo-stability

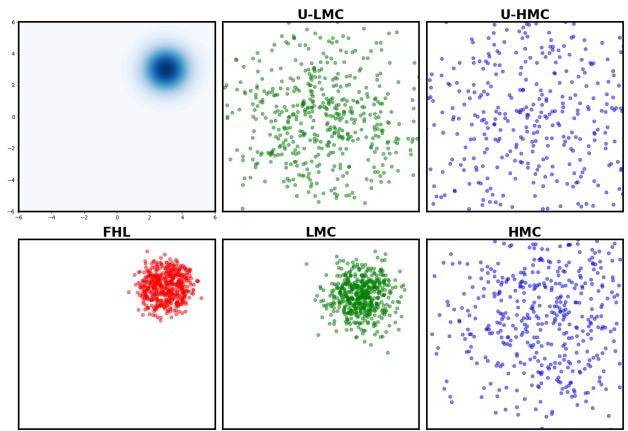

Figure 5: Plot of $N = 256$ particles of $X_T$ on $d = 2$ starting from random initialization $\mathcal{N}(0, 4 \cdot I)$. The target distribution is $\mathcal{N}([1, 1], I)$. Energy State corresponds to the target density $\pi$. The baseline methods U-LMC,LMC,U-HMC,HMC and our proposed method FHL generate $X_T$ after $T = 1000$ steps. There are no gradient flows and the samplers are only able to sample by the energy information.

This section highlights the phenomenon showcased in Figure 3. Here, particles can become ensnared by gradient flows and be coerced into pseudo-stable regions. Despite the eventual recovery of the correct distribution by the sampling method, the convergence process can be exceptionally sluggish.

To elaborate, we examine a scenario where the samplers solely depend on gradients from $\nabla \log Q$, while the energy function $P$ remains deliberately undisclosed. The distributions $P$ and $Q$ are:

- $Q \sim \frac{1}{4} \left[ \mathcal{N}\left(\mu_1, I\right) + \mathcal{N}\left(\mu_2, I\right) + \mathcal{N}\left(\mu_3, I\right) + \mathcal{N}\left(\mu_4, I\right) \right]$
- $P \sim \frac{1}{2} \left[ \mathcal{N}\left(\mu_1, I\right) + \mathcal{N}\left(\mu_2, I\right) \right]$

where $\mu_1 = [-2, 0]$, $\mu_2 = [2, 0]$, $\mu_3 = [0, 2]$ and $\mu_4 = [0, -2]$.

From Figure 6 we can see that FTH does not only capture the modes more quickly compared to the other methods but also successfully get out of the trap of the pseudo-stable regions.

Our study addresses the challenges of instability, metastability, and pseudo-stability, demonstrating that the FTH method consistently outperforms other approaches across various scenarios. Through illustrative experiments, we show that FTH rapidly captures modes and effectively escapes pseudo-stable regions, even when gradients are entirely absent. This superior performance is attributed to FTH's unique leader pulling strategy, which directs particles efficiently to high-probability density regions, thereby avoiding unproductive exploration in low-probability areas.

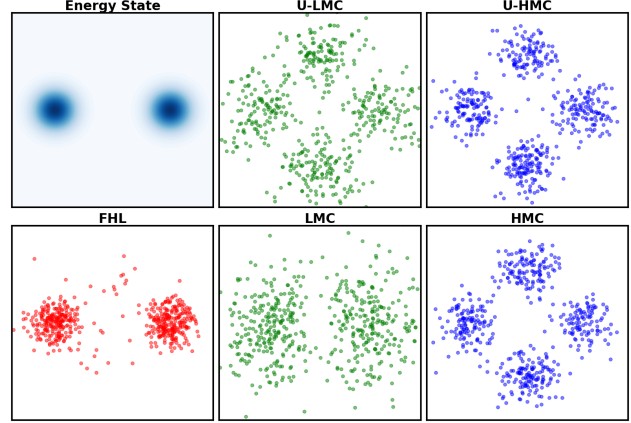

Figure 6: Plot of $N = 256$ particles of $X_T$ for a 2-mode Gaussian mixture model on $d = 2$ starting from random initialization $\mathcal{N}(0, 4 \cdot I)$. Energy State corresponds to the target density $\pi$. The baseline methods U-LMC,LMC,U-HMC,HMC and our proposed method FHL generate $X_T$ after $T = 200$ steps.

In the following section, we will illustrate the advantages of FTH in more general applications, particularly for energy-based (score-matching) models.

## 5.2 Energy-Based Generative Model

Energy-based models (EBMs) offer significant advantages for sampling because they naturally provide energy information that can be utilized to guide the sampling process. In an EBM, the energy function assigns lower energy values to more probable configurations, enabling the sampler to more effectively navigate the probability landscape and generate high-quality samples. This makes EBMs a powerful tool in scenarios where precise sampling is essential.

We investigate a scenario where energy functions guide the sampling process. We use the generative model outlined in [16] and adopt a conditional generation method that leverages classifier-derived gradients for sampling. The classifier's output is considered as the energy for guided sampling. The common classification tasks involving $C$ classes are often solved by using a neural network $f_\theta : \mathbb{R}^d \to \mathbb{R}^C$, which maps each input data point $x \in \mathbb{R}^d$ to $C$-categorical outputs. The output are then used to define a categorical distribution of class $y$ through a *softmax* function:

$$p_\theta(y \mid x) = \frac{\exp(f_\theta(x)[y])}{\sum_{y'} \exp(f_\theta(x)[y'])},$$

where $f_\theta(x)[y]$ represents the $y$-th component of $f_\theta(x)$, corresponding to the logit for class $y$. Once the classifier is trained, $p'_\theta(y \mid x) = \exp(f_\theta(x)[y])$ could be used to sample for a specific class $y$.

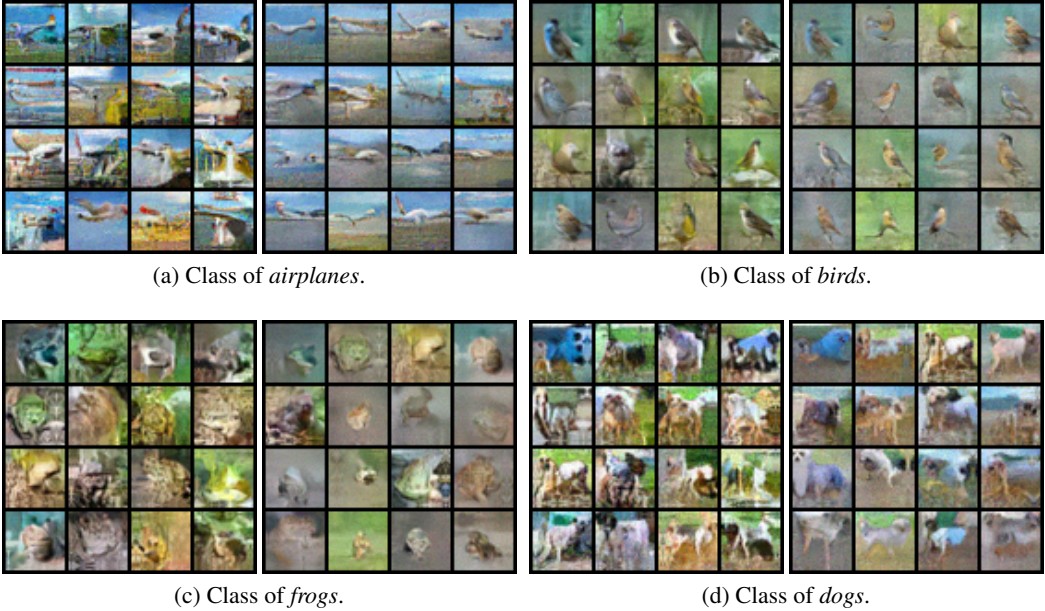

(a) Class of *airplanes*.        (b) Class of *birds*.

(c) Class of *frogs*.        (d) Class of *dogs*.

Figure 7: Sample from joint energy model by different classes (**Left:** HMC; **Right:** FTH).

We compare FTH with the standard HMC method using a limited number of sampling steps, consistently accepting new proposals based on the potential energy during sampling. It is evident that FTH produces higher-quality images than HMC. Additionally, our experiments reveal that FTH tends to generate sharper images compared to the other method. This can be attributed to the assumption that the classifier focuses on the object's features rather than the entire image. As a result, when the prediction probability is high, the features that increase confidence become more prominent, while unrelated background elements are filtered out.

## 5.3 Energy-Based Score-Matching Models

As indicated in [12], when two diffusion models are combined into a product model $q^{\mathrm{prod}}(x) \propto q^1(x)q^2(x)$, problems can arise if the model reversing the diffusion uses a score estimate derived by simply adding the score estimates of the two independent models. We use energy-based score-matching models to illustrate this issue. It is important to note that such inconsistencies typically involve the composition of two or more diffusion models.

### 5.3.1 Synthetic Dataset

We first show an example of composing two distributions $p_1(x)$ and $p_2(x)$, as illustrated in the left column of Figure 8. The results show that FTH demonstrates a strong ability to converge to the correct composition, with less particles fall out of the high-density region compared to others.

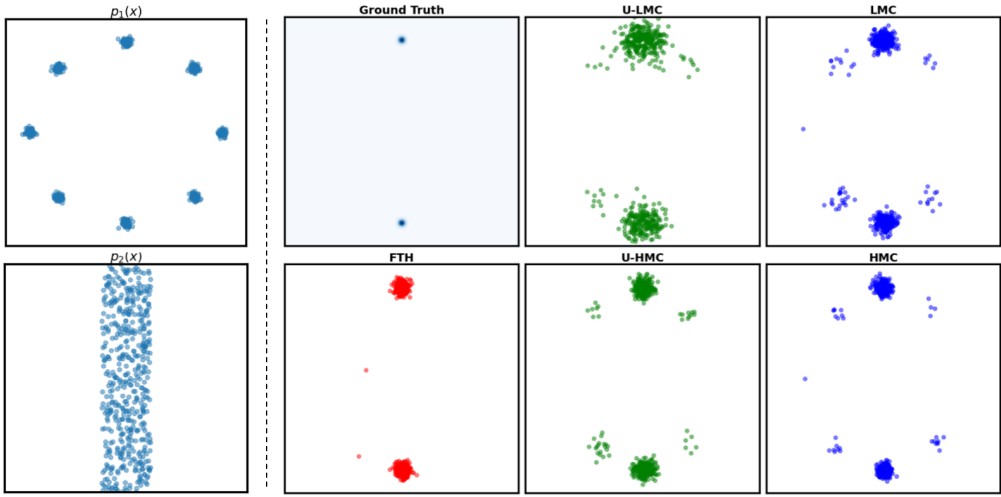

Figure 8: Compose sampling with DDPM.

### 5.3.2 CLEVR Dataset

We use CLEVR dataset from [23] for our generation and sampling tasks. The energy model is adopted from [12], and we employ different samplers for generation. The dataset includes three classes: *cube*, *sphere*, and *cylinder*. We explore scenarios where we first sample from only *one* category and then from *two* categories.

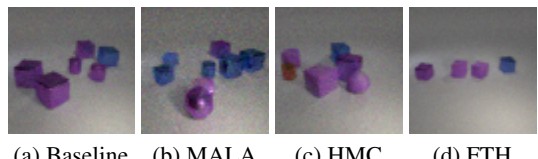

(a) Baseline    (b) MALA    (c) HMC    (d) FTH

Figure 9: Generation of *cube*. The zoomed images could be found at Figure 19.

In the first experiment, there is no composition of models. As depicted in Figure 9, it is evident that FTH effectively generates the desired image without any extraneous shapes, whereas both MALA and HMC generate additional shapes.

In the second experiment, we combine two independent diffusion models, each trained separately to generate *sphere* and *cylinder*. As shown in Figure 10, it is clear that FTH excels at producing high-quality images with almost no over-

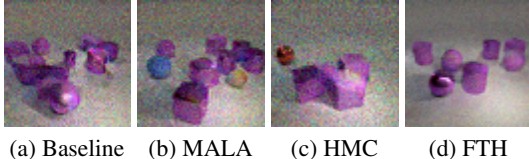

(a) Baseline    (b) MALA    (c) HMC    (d) FTH

Figure 10: Generation of *sphere* and *cylinder*. The zoomed images could be found at Figure 20.

lapping between objects, accurately rendering the intended shapes in a pristine manner. In contrast, the other methods generate the undesired shape *cube*. Additionally, FTH exhibits less noise, indicating greater stability for sampling.

## 6 Conclusion

In this study, we first recognize the significance of incorporating zeroth-order information into the sampling process, highlighting the common limitations faced by conventional sampling methods. These limitations include unstable sampling outcomes frequently associated with energy-based score-matching models, the potential metastability arising from the multi-modal nature of the energy function, and errors in gradient computation stemming from the complex structure of the compositional distribution. Subsequently, we present an innovative approach that leverages parallel HMC sampling to address the issues. Building upon HMC, we incorporate energy modulation techniques to enhance the sampling process. Through this approach, our method is able to systematically reduce the potential energy, leading to substantial advantages in practical implementations of sampling.

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

# Follow Hamiltonian Leader: An Efficient Energy-Guided Sampling Method (Supplementary Material)

## A  Additional Discussion for Section 3

### A.1  Instability & Metastability

We now approach this problem from an optimization perspective. There is a strong connection between optimization and sampling, particularly through the principle of simulated annealing [24], which demonstrates how sampling methods can be transformed into optimization techniques.

With a slight abuse of notation, we consider the following objective function:

$$U(x) = x[1]^2 + 0.01 \cdot x[2]^2,$$

where $x \in \mathbb{R}^2$ and $x[i]$ denotes the $i^{th}$ dimension of $x$. This is a 2-dimensional optimization problem with a condition number of 100, indicating it is somewhat ill-conditioned.

For $n$ particles, the objective function is:

$$U_e(x^1, \cdots, x^n; x^l) = \sum_{i=1}^{n} U(x^i) + \frac{\rho}{2} \cdot \|x^i - x^l\|_2^2,$$

where we set $\rho = 0.1$. We initialize $x^1 = (2, 2)$ and $x^2 = (-1, -3)$ respectively, and optimize the objective function using the gradient descent method. Note that when $n = 1$, this method reduces to vanilla gradient descent, while $n = 2$ incorporates our leader-pulling scheme.

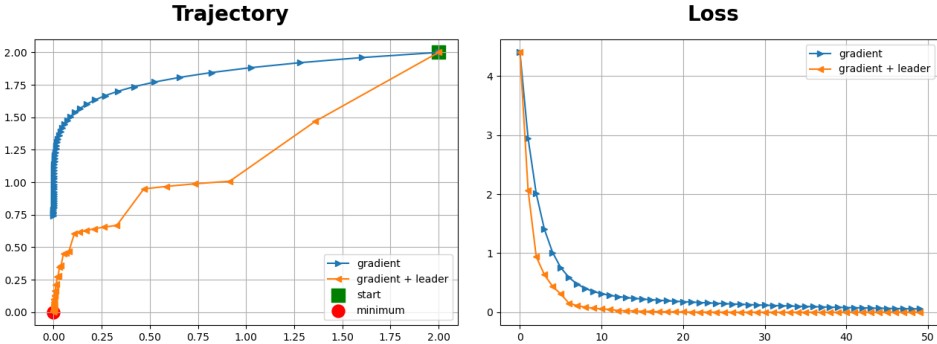

Figure 11: $U(x^1)$ with gradient descent method [6]. The learning rate is set to 0.1.

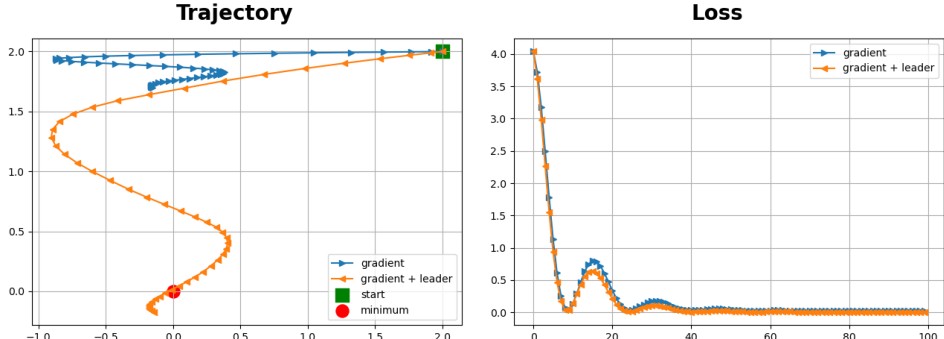

Figure 12: $U(x^1)$ with heavy-ball method [31]. The learning rate and momentum are set to $0.02$ and $0.9$ respectively.

From Figure 11, we can see that incorporating the leader-pulling scheme helps improve convergence. This demonstrates that the leader-pulling scheme can address the issue of instability in optimization. However, we also observe that a carefully chosen leader is usually required for our method, which we will leave for future discussion.

Furthermore, as shown in Figure 12, the particle using the leader-pulling scheme explores much further compared to the vanilla heavy-ball method. This outcome is expected, as we want the method to enhance exploration and thereby resolve the metastability issue.

## A.2 Pseudo Stability

These challenges are commonly encountered when sampling from compositional models, particularly when one of the distributions is a piecewise-constant distribution with its gradients are zero almost everywhere in its domain. To illustrate this, consider the example $\pi(x) \propto \pi_1(x) \cdot \pi_2(x)$. Here we consider $\partial \log \pi_2$ equals to zero everywhere.

It's worth noting that while combining distributions in their logarithmic forms is straightforward, which leads to $\log \pi(x) = \log \pi_1(x) + \log \pi_2(x) +$ constant , omitting the constant $\log \pi(x)$ can be readily derived from the individual $\log \pi_1(x)$ and $\log \pi_2(x)$. However, the composition of their gradients becomes problematic, as the computation of the sub-gradient $\partial_x \log \pi(x) \neq \nabla_x \log \pi_1(x) + \partial_x \log \pi_2(x)$ in general due to the use of automatic differentiation in machine learning [4].

In this section, we focus on the disparity between gradient and energy in the context of combining two distributions as indicated in Section 3.

We analyze a composite probability distribution structured as $\pi(x) \propto \pi_1(x) \cdot \pi_2(x)$, leading to the construction of two specific distributions:

- The first distribution, $\pi_1(x)$, is given by:

$$\pi_1(x) = \frac{1}{|X|} \sum_{\mu \in X} \mathcal{N}(\mu, \sigma^2 I),$$

  where $|X|$ represents the cardinality of the set $X$, indicating the total number of elements in $X$.

- The second distribution $\pi_2(x)$, is defined as

$$\pi_2(x) = \frac{\mathbb{1}_{x \in \Omega_Y}}{\text{Vol}(\Omega_Y)},$$

  with $\Omega_Y$ being the set where $\Omega_Y = \{x \mid d(x, Y) < \epsilon\}$. In this context, the distance metric $d$ is specified by $d(x, Y) = \arg\min_{y \in Y} \|x - y\|_2$, indicating the minimum Euclidean distance from $x$ to any point in the set $Y$.

Observe that $\pi_1$ constitutes a smooth distribution, whereas $\pi_2$ is a piecewise-constant distribution. Consequently, for $\pi_2$, the gradients are zero almost everywhere. When we consider the expression $\nabla_x \log \pi_1(x) + \partial_x \log \pi_2(x)$, it could be simplified to $\nabla_x \log \pi_1(x)$, which is not equivalent to $\partial_x \log \pi(x)$ in general.

In the subsequent subsections, we present two motivating examples: one in a low-dimensional setting and the other in a high-dimensional context. Throughout these experiments, we set $\sigma^2 = 0.002$ and $\epsilon = 0.2$. In this section, the outcomes of U-LMC and U-HMC are omitted because both techniques succumb to the issue of misleading gradients by nature, causing worse performance.

### A.2.1 Low-dimensional Example

We propose an example inspired from [12] but in a different setting. In this revision, we begin by providing a more specific definition for two distributions. For the first distribution $\pi_1$, we define:

$$X = \{ ( \cos(2\pi i/8), \ \sin(2\pi i/8) ) | \ i = 1, 2, \ldots, 8 \},$$

and for the second distribution $\pi_2$, we specify:

$$Y = \{ ( \cos(2\pi i/8), \ \sin(2\pi i/8) ) | \ i = 2, 4, 6, 8 \}.$$

It's important to note that, by definition, $Y$ is a subset of $X$.

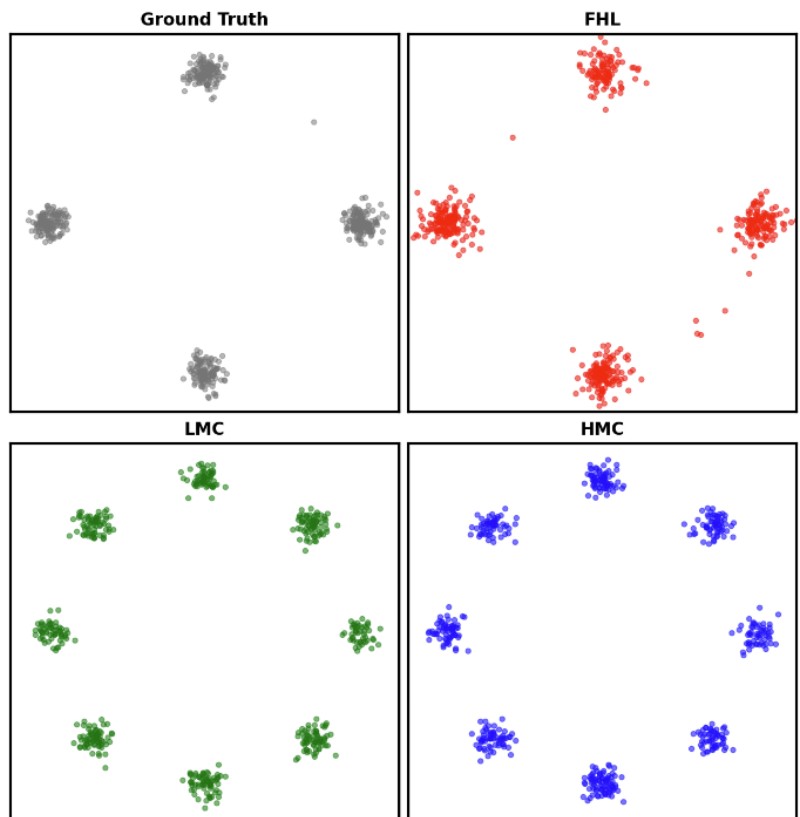

Figure 13: Plot of $N = 512$ particles of $X_T$ for a 4-mode compositional Gaussian mixture model $\pi \propto \pi_1 \cdot \pi_2$ on $d = 2$. We sample by gradient $\nabla \log \pi_1$ and energy $\pi_1 \cdot \pi_2$. The baseline methods LMC, HMC and our proposed method FHL generate $X_T$ after $T = 4000$ steps, using the initial particles $X_0 = \{x_0^i\}$ with $x_0^i$ sampled from a common distribution.

We perform a comparative study of our methods against established benchmarks, and the visual representations of this comparison can be found in Figure 13. Notably, among the compared methods, FTH distinguishes itself due to its outstanding performance, mainly attributed to its precise adjustment of particle positions. The comparative results highlight that the baseline methods often exhibit the tendency to erroneously converge towards incorrect modes due to the misleading gradients. Although rejection steps of HMC and LMC might mitigate incorrect sampling, particles initialized near high-energy modes struggle to escape this erroneous attraction by misleading gradients.

### A.2.2 High-dimensional Example

We then present a case study in which we generate examples from a particular category within the Fashion MNIST dataset [43]. In this experiment, we select a total of 200 images, with 100 images

from the *coat* category and another 100 from *trouser* category. We denote the sets of data points from the *coat* and *trouser* categories as $X_{\text{coat}}$ and $X_{\text{trouser}}$ respectively. Furthermore, we define $X$ as the union of $X_{\text{coat}}$ and $X_{\text{trouser}}$, and $Y$ is set to $X_{\text{coat}}$ in this case.

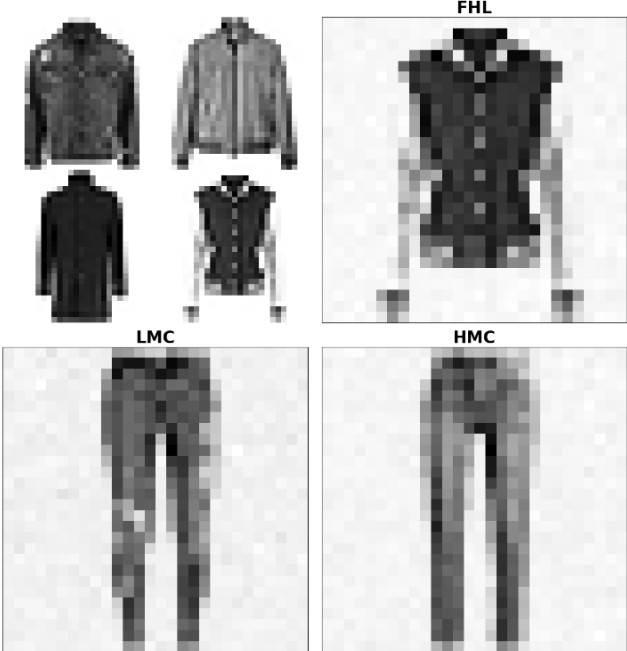

Figure 14: Sample from a 100-mode compositional Gaussian mixture model $\pi \propto \pi_1 \cdot \pi_2$ on $d = 784$, where each mode corresponds to a clean image from *coat* category. We sample by gradient $\nabla \log \pi_1$ and energy $\pi_1 \cdot \pi_2$. For each method, we plot the smallest-energy particle (in terms of $U(x)$ among all particles in $X_T$). The correct samples are displayed in the upper-left corner.

To increase the difficulty of the sampling task, we initially position each particle at the mean location of $X_{\text{trouser}}$. The outcomes of the sampling are depicted in Figure 15. This setup showcases the FHL method's ability to accurately target and sample from the specified *coat* category, in contrast to baseline methods that undesirably draw samples from the *trouser* category.

# B Supplementary Experiment

## B.1 Experiment Setup

**In Section 5.2**, we utilize a pre-trained classifier available on the public GitHub repository at `https://github.com/wgrathwohl/JEM`. This classifier is a WideResNet model [45] with a depth of 28 and a width of 2.

We use a technique called one-step HMC [5] and thus the momentum gets refreshed for each step. More specifically, for both FTH and HMC we set the momentum damping factor to 0.9 and the mass matrix as $0.004^2 \cdot I$. We take step size as $\eta = 0.2$. Since the mass matrix is set to a relatively small value which easily causes the instability of training, we always accept the proposed states based on the potential energy and ignore the kinetic energy.

**In Section 5.3**, we mainly adapted the codes and models from `https://github.com/yilundu/reduce_reuse_recycle`.

For Section 5.3.1, we initially train a 4-layer ResNet as the energy-based score-matching model on $p_1$ and $p_2$ independently. During the sampling process, we combine these models. We employ step sizes $\eta = \{0.002, 0.0002, 0.005, 0.0005\}$ for all methods and the number of leapfrog steps $L = \{4, 8\}$ for HMC-type methods.

For Section 5.3.2, we utilize a U-net architecture [33] as the energy-based score-matching model. This architecture is directly obtained from a pre-trained model available at . For sampling, we use step sizes $\eta = \{0.01, 0.035, 0.05, 0.1, 0.2\}$ for all methods and set the number of leapfrog steps $L = 4$ for HMC-type methods.

 **B.2 Additional Results**

 **B.2.1 Additional Images for Section 5.2**

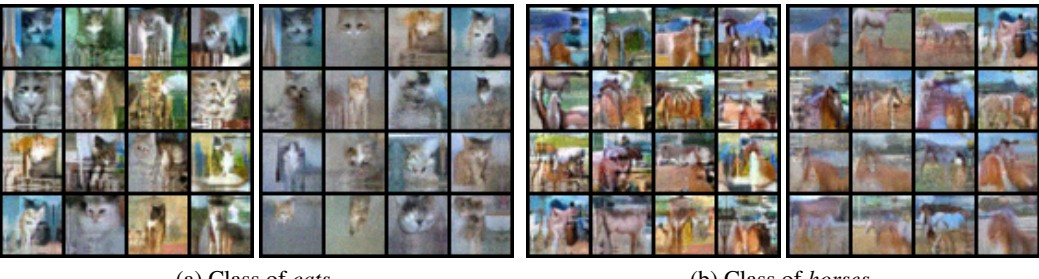

(a) Class of *cats*.          (b) Class of *horses*.

Figure 15: Sample from joint energy model by different classes (**Left:** HMC; **Right:** FTH).

 **B.2.2 Additional Images for Section 5.3.2**

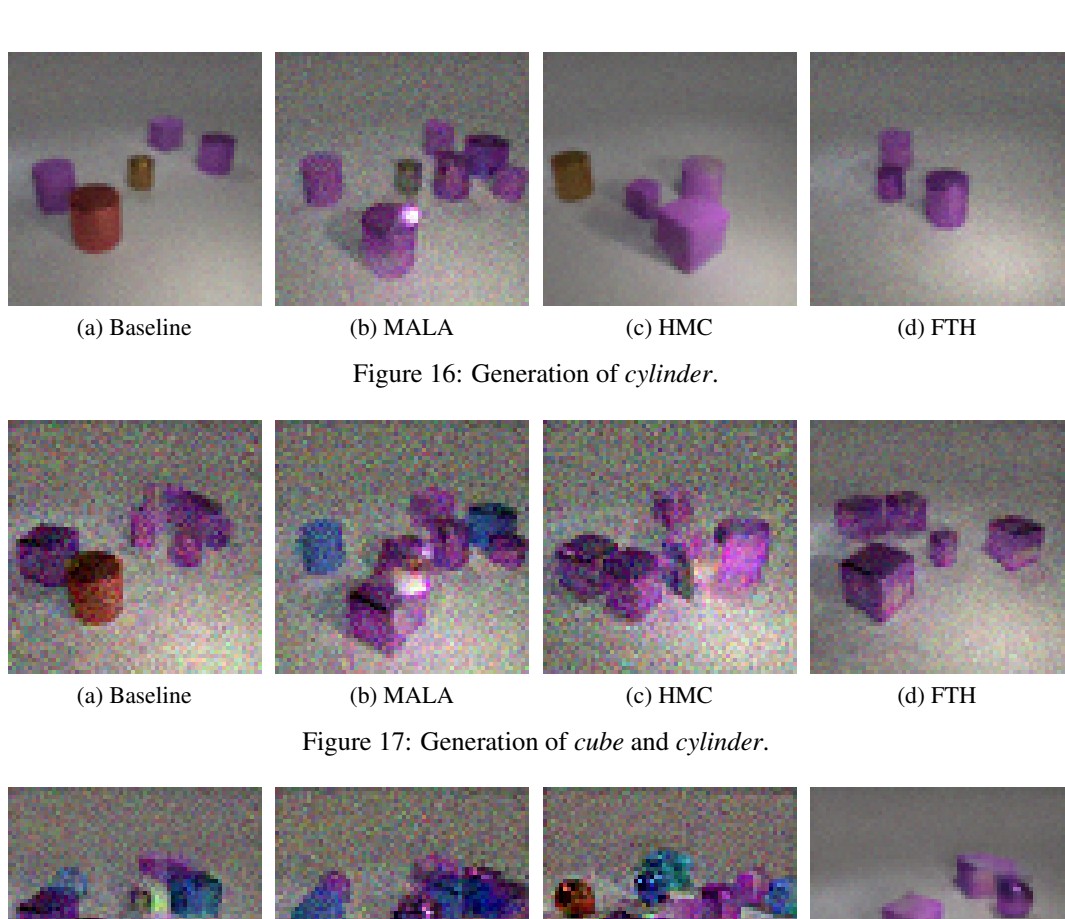

(a) Baseline      (b) MALA      (c) HMC      (d) FTH

Figure 16: Generation of *cylinder*.

(a) Baseline      (b) MALA      (c) HMC      (d) FTH

Figure 17: Generation of *cube* and *cylinder*.

(a) Baseline      (b) MALA      (c) HMC      (d) FTH

Figure 18: Generation of *cube* and *sphere*.

 **B.2.3 Zoomed Images for Section 5.3.2**

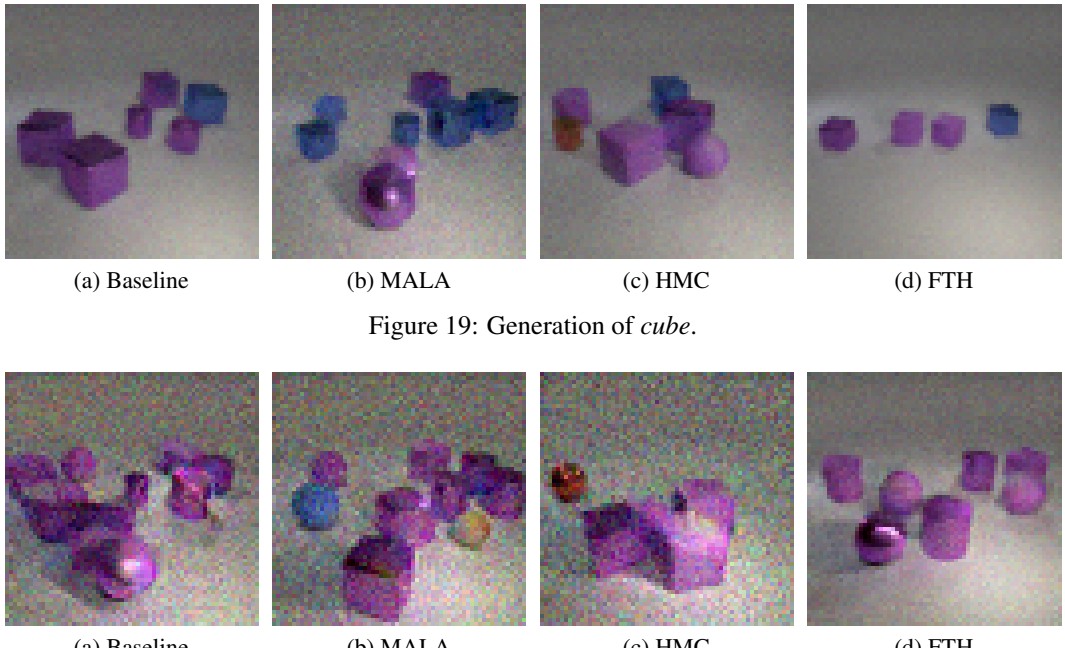

(a) Baseline      (b) MALA      (c) HMC      (d) FTH

Figure 19: Generation of *cube*.

(a) Baseline      (b) MALA      (c) HMC      (d) FTH

Figure 20: Generation of *sphere* and *cylinder*.

## C  Supplementary Theorem

We now consider a scenario where the leader becomes corrupted, meaning the corrupted leader always reports an unreasonably low energy but it is not actually in the lowest-energy position. In this situation, the particles are optimizing a biased objective function. For simplicity, we consider a $d$-dimensional Gaussian distribution $p \sim e^{-U(x)}$ and its modification $q \sim e^{-\psi(x)}$ with $\psi(x) = U(x) + \frac{\lambda}{2}\|x - z\|^2$. We will analyze the Wasserstein distance between $p$ and $q$ for a fixed $z \in \mathbb{R}^d$ as a function of $\lambda > 0$. We will demonstrate that even though we sample from the distribution $q$ instead of $p$, the bias of the sampler (i.e., the distance between $p$ and $q$) can be controlled by $\lambda$ and vanishes as $\lambda \to 0$.

*Assumption* 1. $U : \mathbb{R}^d \to \mathbb{R}$ is $M$-Lipschitz-differentiable, i.e. $\forall x, y \in \mathbb{R}^d$,

$$U(y) \leq U(x) + \nabla U(x)^T(y - x) + \frac{M}{2}\|y - x\|^2,$$

and $U$ is $m$-Strongly-convex, i.e. $\forall x, y \in \mathbb{R}^d$,

$$U(y) \geq U(x) + \nabla U(x)^T(y - x) + \frac{m}{2}\|y - x\|^2.$$

*Theorem* 1. Let $U$ be the negative logarithmic probability density function of a $d$-dimensional Gaussian distribution, which satisfies Assumption 1. Let us define the function $\psi(x)$ as $\psi(x) = U(x) + \frac{\lambda}{2}\|x - z\|^2$. Given this setup, the Wasserstein-2 distance between the modified Boltzmann distribution $q$, characterized by $q \sim e^{-\psi(x)}$, and the original Gaussian distribution $p$, denoted as $p \sim e^{-U(x)}$, can be bounded as:

$$W_2(p, q)^2 \leq \frac{\lambda^2\|\Sigma\|}{I + \lambda\|\Sigma\|}\|z - x^*\|^2 + d\|\Sigma\| \cdot \left(1 - \frac{1}{\sqrt{\lambda\|\Sigma\| + 1}}\right)^2$$

where $\|\cdot\|$ represents the matrix norm. Obviously, $W_2(p, q) \to 0$ when $\lambda \to 0$.

*Proof.* By definition, $U$ is $m$-strongly convex since

$$U(x) = -\log\left[(2\pi)^{-k/2}\det(\Sigma)^{-1/2}\exp\left(-\frac{1}{2}(x-x^*)^T\Sigma^{-1}(x-x^*)\right)\right]$$

$$= \frac{k}{2}\log(2\pi) + \frac{1}{2}\log\det(\Sigma) + \frac{1}{2}(x-x^*)^T\Sigma^{-1}(x-x^*).$$

The $m$ corresponds to the smallest eigenvalue of $\Sigma^{-1}$ which is therefore $1/\|\Sigma\|$. Then

$$\psi(x) = \frac{k}{2}\log(2\pi) + \frac{1}{2}\log\det(\Sigma) + \frac{1}{2}(x-x^*)^T\Sigma^{-1}(x-x^*) + \frac{\lambda}{2}(x-z)^T(x-z)$$

$$= \frac{1}{2}\left[x^T(\Sigma^{-1}+\lambda I)x - 2x^T(\Sigma^{-1}x^* + \lambda z)\right] + \text{constant}$$

$$= \frac{1}{2}\left[(x-(\Sigma^{-1}+\lambda I)^{-1}(\Sigma^{-1}x^*+\lambda z))^T(\Sigma^{-1}+\lambda I)(x-(\Sigma^{-1}+\lambda I)^{-1}(\Sigma^{-1}x^*+\lambda z))\right] + \text{constant}$$

The last equation was done by completing the square. Thus the new distribution is still a Gaussian distribution, represented as

$$q \sim \mathcal{N}\left((\Sigma^{-1}+\lambda I)^{-1}(\Sigma^{-1}x^*+\lambda z)), (\Sigma^{-1}+\lambda I)^{-1}\right).$$

Consequently, the Wasserstein-2 distance can be determined as follows:

$$W_2(p,q)^2 = \|\mu_p - \mu_q\|^2 + \text{Tr}\left(\Sigma_p + \Sigma_q - 2(\Sigma_p^{1/2}\Sigma_q\Sigma_p^{1/2})^{1/2}\right)$$

In our case $\mu_p = x^*, \mu_q = (\Sigma^{-1}+\lambda I)^{-1}(\Sigma^{-1}x^*+\lambda z), \Sigma_q = \Sigma, \Sigma_p = (\Sigma^{-1}+\lambda I)^{-1}$. Since $\Sigma_q$ and $\Sigma_p$ can be jointly diagonalized by some orthonormal basis $T$,

$$\Sigma_q\Sigma_p = TD_qT^{-1}TD_pT^{-1} = TD_pD_qT^{-1} = TD_pT^{-1}TD_qT^{-1} = \Sigma_p\Sigma_q,$$

thus $\Sigma_q$ and $\Sigma_p$ commute. We can simplify the Wasserstein distance to

$$W_2(p,q)^2 = \|\mu_p - \mu_q\|^2 + \|\Sigma_p^{1/2} - \Sigma_q^{1/2}\|_F^2.$$

Then

$$W_2(p,q)^2 = \|(\Sigma^{-1}+\lambda I)^{-1}(\Sigma^{-1}x^*+\lambda z) - x^*\|^2 + \|\Sigma^{1/2} - (\Sigma^{-1}+\lambda I)^{-1/2}\|_F^2.$$

Now we bound the first and second term independently. The first term is a direct conclusion from Theorem 15 in [41],

$$\|(\Sigma^{-1}+\lambda I)^{-1}(\Sigma^{-1}x^*+\lambda z) - x^*\|^2 \leq \frac{\lambda^2}{m(m+\lambda)}\|z-x^*\|^2, \quad \text{where } m = \|\Sigma^{-1}\|,$$

For the second term. We denote $i^{th}$ eigenvalue of matrix $\Sigma$ as $\sigma_i$, then $\|\Sigma\| = \max_i \sigma^i$, such that

$$\|\Sigma^{1/2} - (\Sigma^{-1}+\lambda I)^{-1/2}\|_F^2 = \sum_{i\leq d}[\sigma_i^{1/2} - (\sigma_i^{-1}+\lambda)^{-1/2}]^2$$

$$= \sum_{i\leq d}[\sqrt{\sigma_i}\cdot(1 - \frac{1}{\sqrt{\lambda\sigma_i+1}})]^2$$

$$\leq d\|\Sigma\|\cdot\left(1 - \frac{1}{\sqrt{\lambda\|\Sigma\|+1}}\right)^2$$

Thus, by combing the two terms together, the total Wassertein distance is bounded by

$$W_2(p,q)^2 \leq \frac{\lambda^2}{m(m+\lambda)}\|z-x^*\|^2 + dM\cdot\left(1 - \frac{1}{\sqrt{\lambda M+1}}\right)^2$$

$\square$

