# OpenReview forum: "Follow Hamiltonian Leader: An Efficient Energy-Guided Sampling Method"
_NeurIPS.cc/2024/Conference — Submitted to NeurIPS 2024_

### Official Review · Reviewer_M1Sx · 2024-07-08

**Soundness:** 3
**Presentation:** 3
**Contribution:** 3
**Rating:** 6
**Confidence:** 4

**Summary:**

This paper presents a novel parallel sampling method named "Follow Hamiltonian Leader" (FHL) designed to address sampling challenges by leveraging zeroth-order information, particularly when first-order data is unreliable or unavailable. The method incorporates a leader-guiding mechanism to enhance the efficiency and effectiveness of the sampling process. Experimental results indicate that FHL significantly improves the exploration of target distributions and outperforms traditional sampling techniques, especially in scenarios involving corrupted gradients.

**Strengths:**

1. Innovative combination of zeroth and first-order information.
2. The effectiveness of the method is demonstrated in multiple task scenarios.
3. Theoretical analysis and prove are sufficient.

**Weaknesses:**

1. Is there any quantitative experiments like evaluating FID and IS on cifar10 datasets and I think it's more compelling whether a novel sampling methods combined with generative models can be used on image datasets with more complex distributions.
2. Lack of experiment of OOD in combination with EBMs or score-based models to valid the stability during sampling with proposed method.

**Questions:**

Why is E(x, \tilde x) defined in this way. And is there any relation between extra elastic energy and local entropy [1].


[1] P. Chaudhari, A. Choromanska, S. Soatto, Y. LeCun, C. Baldassi, C. Borgs, J. T. Chayes, L. Sagun, and R. Zecchina. Entropy-SGD: Biasing gradient descent into wide valleys. In ICLR, 2017.

**Limitations:**

1. Limited exploration of integration with other advanced MCMC methods.
2. Lack of quantitative experiments to demonstrate the advantage of proposed sampling method compared with other methods.

---

> ### Author Rebuttal · Authors · 2024-08-07
>
> **Rebuttal**
>
> Thank you for reviewing our paper and providing insightful feedback. We appreciate your positive remarks and constructive criticism, which will help us improve our work. Below, we address your comments and questions:
>
> 1. **Quantitative Experiments on CIFAR-10:**
>    We understand your concern about the absence of quantitative experiments on complex image datasets. To address this, we have conducted experiments to evaluate the performance of FHL on the CIFAR-10 dataset, and the results are included in the overall response. These findings will be incorporated into the revised manuscript to highlight the method's applicability to complex distributions.
>
> 2. **Out-of-Distribution (OOD) Experiments with EBMs or Score-Based Models:**
>    We apologize for any confusion regarding out-of-distribution experiments in our context. Our method is fundamentally a sampling technique designed to accurately generate the target distribution, whereas out-of-distribution experiments are typically performed in classification or detection tasks.
>
>    However, as an alternative, we had conducted experiments in scenarios where sampling is challenging, such as sampling from poorly-conditioned functions, as shown in Figure 4. These experiments aim to validate the stability and robustness of FHL during sampling. We are glad to extend the results in the revised paper to provide a more comprehensive analysis of the method's capabilities.
>
> **Discussions**
>
> 1. **Integration with Advanced MCMC Methods:**
>    We recognize that our algorithm could benefit from integration with other advanced MCMC methods. In fact, our method is orthogonal to these techniques and can be combined with methods such as LMC or parallel tempering. However, due to space and time constraints, our current version focuses solely on comparison and integration with HMC. We will extend our work to include integration with other advanced MCMC methods in the future.
>
> 2. **Quantitative Comparison with Other Methods:**
>    As suggested, we have included quantitative experimental results for CIFAR10 in the overall response.
>
>
> **Question**
>
>    - **Entropy-SGD**
>
>      The key concept of Entropy-SGD is its incorporation of Bayesian optimization elements. Unlike other optimization methods such as Adam or Momentum SGD, Entropy-SGD optimize over a local area (referred to as "local entropy") instead of searching for an isolated optimal point.
>
>      $$
>      F(x, \gamma) = \log \int_{x' \in \mathbb{R}^n} \exp\left( -f(x') - \frac{\gamma}{2} ||x - x'||_2^2 \right)  dx'.
>      $$
>
>      The extra term $E(x, x') = \frac{\gamma}{2} ||x-x'||^2_2$ in Entropy-SGD controls the range over which it seeks valleys of specific widths. Another way to write the objective used in the Entropy-SGD (see Equation 3, [2]) is:
>      $$
>      f_\gamma(x) := -\log \left( G_\gamma * e^{-f(x)} \right); G_\gamma = (2\pi\gamma)^{-N/2} \exp\left( -\frac{||x||^2}{2\gamma} \right)
>      $$
>
>      The objective above shows that Entropy-SGD optimizes a modified, smoother energy landscape, which can be viewed as a convolution of the original energy landscape with a Gaussian kernel. This approach aids in identifying wider valleys.
>
>    - **PARLE: The Parallel Variant of Entropy-SGD**
>      An extension of Entropy-SGD, known as PARLE [2], essentially parallelizes the Entropy-SGD algorithm. This method optimizes the following objective function, which reduces to Entropy-SGD when $n=1$:
>
>      $$
>      \arg \min_{x, x^1, \ldots, x^n} \sum_{a=1}^{n} f_\gamma(x^a) + \frac{1}{2\rho} || x^a - x ||^2.
>      $$
>
>      By deriving this expression, it becomes evident that PARLE encourages particles to sample around $\bar{x}$, where $\bar{x} = \frac{1}{n} \sum_{a=1}^{n} x^a$ is the average of the particles.
>
>    - **Connection and Comparison for FHL**
>
>      Both Entropy-SGD and PARLE are optimization methods that ultimately produce a final point instead of a Markov chain. In contrast, FHL is specifically designed as a sampling method. Furthermore, there are other similarities and differences between FHL and Entropy-SGD/PARLE:
>
>      - **(Similarity) Exploring the Landscape:** All of three approaches allow to explore the function landscape around the anchor point (Entropy-SGD: the average of the SGD trajectory of a single particle; PARLE: the average of multiple particles; FHL: the leader of the particles).
>
>      - **(Difference) Zeroth-Order Optimization:** FHL leverage zeroth-order information, and uses the leader as the anchor point instead of the average of particles. Averaging is commonly used in optimization techniques to reduce variance. However, in sampling tasks, noise is typically not introduced, and pulling towards the average may degrade convergence performance. Therefore, FHL could be expected to perform better.
>
>    > [1] P. Chaudhari, A. Choromanska, S. Soatto, Y. LeCun, C. Baldassi, C. Borgs, J. T. Chayes, L. Sagun, and R. Zecchina. Entropy-SGD: Biasing gradient descent into wide valleys. In ICLR, 2017.
>    >
>    > [2] Chaudhari, Pratik, et al. "Parle: parallelizing stochastic gradient descent." arXiv preprint arXiv:1707.00424 (2017).
>
>
> ---
> Thank you again for your valuable feedback. We are confident that the additional results and planned revisions will address your concerns and enhance the overall quality and impact of our work.

---

> > ### Comment · Reviewer_M1Sx · 2024-08-11
> >
> > Thank you for your detailed response and additional experiments. I have also read the reviews from other reviewers as well as the corresponding reply. I have increased my score.

---

### Official Review · Reviewer_N2ad · 2024-07-12

**Soundness:** 2
**Presentation:** 3
**Contribution:** 2
**Rating:** 4
**Confidence:** 2

**Summary:**

This paper introduces an interesting parallel sampling method that leverages zeroth-order information to address challenges in sampling from probability distributions, particularly when first-order data is unreliable or unavailable. The method incorporates a leader-guiding mechanism, enhancing efficiency and effectiveness by connecting multiple sampling instances through a selected leader. The proposed method, named Follow Hamiltonian Leader (FHL), extends the Hamiltonian Monte Carlo (HMC) framework by concurrently running multiple replicas at different energy levels and combining both zeroth and first-order information from various chains. Experimental results demonstrate that FHL significantly improves the exploration of target distributions and produces higher-quality outcomes compared to traditional sampling techniques, showing resilience against corrupted gradients and excelling in scenarios characterized by instability, metastability, and pseudo-stability.

**Strengths:**

- The proposed Follow Hamiltonian Leader (FHL) method markedly improves the efficiency and effectiveness of sampling processes, significantly expediting the exploration of target distributions and producing superior quality outcomes compared to traditional sampling techniques.
- FHL demonstrates greater resilience against the detrimental impacts of corrupted gradients by incorporating zeroth-order information. This robustness makes the method particularly valuable in scenarios where first-order information is compromised, ensuring more reliable and accurate sampling.

**Weaknesses:**

- The proposed FHL method involves intricate modifications to the traditional Hamiltonian Monte Carlo framework, such as the leader-guiding mechanism and elastic leapfrog technique, which may increase the complexity of implementation and require significant computational resources.
- The effectiveness of the FHL method heavily relies on the appropriate selection of the leader particle. If the leader is not accurately chosen, it could lead to suboptimal sampling performance, potentially compromising the overall efficiency and accuracy of the method.
- While the paper presents experimental results to demonstrate the efficacy of the FHL method, there is a lack of in-depth theoretical analysis to rigorously establish the convergence properties and performance guarantees of the proposed approach.
- The method’s scalability to high-dimensional problems or extremely large datasets is not thoroughly addressed. The parallel sampling approach may encounter challenges in maintaining efficiency and effectiveness as the dimensionality and size of the data increase.

**Questions:**

See weaknesses above.

**Limitations:**

The authors have not adequately addressed the limitations.

---

> ### Author Rebuttal · Authors · 2024-08-07
>
> Thank you for your detailed review and for highlighting both the strengths and areas for improvement in our paper.
>
> **Explanation**
>
> For most sampling algorithms, the objective is to develop a proposal function $Q(x'|x): \mathbb{R}^d \rightarrow \mathbb{R}^d$ that generates a new sample $x'$ from an existing sample $x$, with the goal of having $x'$ fall into a region of high probability density. Provided the sampling algorithm adheres to the principles of detailed balance and ergodicity, it typically converges to the correct distribution.
>
> Therefore, designing an effective sampling method generally involves improving the proposal at each step. FHL introduces a bias for each particle, guiding them towards more accurate results, specifically aiming for a lower value of the objective function $U(x)$, which aligns with common optimization tasks. There are two potential scenarios where our approach can be beneficial:
>
> 1. **Missing Gradient:** When the gradient is unavailable, the leader can be used as a reference point. This concept can be illustrated with a convex function. Consider a convex function $f$, where $x$ represents our particle and $y$ represents its leader. Then, $f((1-\rho) \cdot x + \rho \cdot y) < (1-\rho)f(x) + \rho f(y) < f(x)$, suggesting that $\rho (y-x)$ is a potential loss-descent step as long as $0 < \rho < 1$. Therefore, even when the gradient vanishes, our method still ensures a decrease in loss.
>
> 2. **Non-Optimal Gradient Direction:** As illustrated in Figure 1, the leader's pulling bias can improve the convergence direction when the gradient descent direction and the Newton descent direction are not aligned. It can be demonstrated that as long as the leader lies within the cone $C = \text{cone}(d_N(x), \theta_x)$, the descent direction could be improved. Here, $d_N(x)$ is the Newton descent direction (red arrow in Figure 1), $d_G(x)$ is the gradient descent direction (blue arrow in Figure 1), and $\theta_x = \theta(d_G(x), d_N(x))$.
>
> We hope this explanation clarifies our contributions and provides a better understanding of the potential of the FHL method.
>
> **Rebuttal**
>
> We have carefully considered your feedback and address your points below:
>
> 1. **Complexity and Computational Resources:**
>    We acknowledge that the FHL method introduces modifications that may increase complexity. However, only one additional hyperparameter, the pulling strength $\lambda$, is specifically introduced, while the number of particles per group can be determined based on available computational resources. FHL is designed for large-scale parallel sampling, supported by two key factors:
>
>    1. The computation can be executed in parallel.
>    2. The communication cost scales ***logarithmically***.
>
>    As the number of particles increases, the total running time is primarily influenced by the communication cost, making FHL highly scalable with the number of particles. Additionally, techniques like lazy communication can be employed to further reduce communication overhead.
>
> 2. **Selection of the Leader Particle:**
>    The leader selection is indeed crucial to the method's performance. However, unlike stochastic optimization, most sampling algorithms require that the zeroth-order information be correct. If this assumption fails, these methods would become ineffective, as the key Metropolis-Hastings step [1] relies on the zeroth-order information to satisfy the detailed balance condition.
>
>    Furthermore, even with suboptimal leader selection, although the convergence rate might be affected, the method should still be able to converge to the correct distribution, as long as the algorithm satisfies the detailed balance condition.
>
>    Even if we are forced to sample from a biased function $\psi(x) = U(x) + \frac{\lambda}{2}||x - z||^2$ instead of the true function $U(x)$, the error can be bounded in some sense. We have provided a theoretical analysis of this in our appendix, Section C.
>
>    > [1] Chib, Siddhartha; Greenberg, Edward (1995). "Understanding the Metropolis–Hastings Algorithm". The American Statistician, 49(4), 327–335.
>
> 3. **Lack of Theoretical Analysis:**
>    We have included a theoretical analysis in the overall response, demonstrating that our algorithm has the correct distribution (to sample from) as its invariant distribution.
>
> 4. **Scalability Concerns:**
>    We have included a large-scale experiment in the overall response, demonstrating that FHL can generate biomolecule conformations despite the high number of degrees of freedom and numerous energy landscape barriers. This sampling experiment is not only extensive but also highly significant for real-world simulations of biomolecules.
>
> ---
>
> Thank you once again for your valuable feedback and insights. We believe the planned revisions and additional analyses will address your concerns and enhance the paper's quality. We appreciate your assessment and are committed to addressing the concerns raised to improve the paper's overall quality and impact.

---

> ### Author Response · Authors · 2024-08-12
>
> Dear Reviewer N2ad,
>
> Thank you for your detailed review and valuable feedback on our paper. We appreciate the opportunity to address your concerns and provide further clarification on our rebuttal.
>
> Addressing Your Concerns:
> * Complexity and Computational Resources: FHL introduces minimal complexity with only one additional hyperparameter. The method is designed for parallel execution, making it scalable as communication costs scale logarithmically. Techniques like lazy communication further reduce overhead.
>
> * Selection of the Leader Particle: While leader selection is crucial, our method remains effective as long as it adheres to the detailed balance condition. Even with suboptimal selection, convergence to the correct distribution is achievable.
>
> * Lack of Theoretical Analysis: We have included a theoretical analysis in our appendix and overall summary (see pdf), demonstrating the correctness of our algorithm's invariant distribution.
>
> * Scalability Concerns: Our large-scale experiments showcase FHL's capability to generate biomolecule conformations effectively, highlighting its significance for real-world applications.
>
> We are committed to improving our paper based on your insights and believe that the planned revisions will address your concerns. Thank you again for your valuable feedback.
>
> Best regards,

---

> > ### Author Response · Authors · 2024-08-13
> >
> > Dear Reviewer N2ad，
> >
> > We kindly request your attention to our rebuttal and the experiments we have included. We value your insights and would appreciate your feedback at your earliest convenience.
> >
> > Receiving your comments sooner rather than later will allow us ample time to make necessary revisions and address any further concerns you may have. We are eager to incorporate your valuable input to enhance the quality of our paper.
> >
> > Thank you for your understanding and cooperation.
> >
> > Best regards,
> > Authors

---

### Official Review · Reviewer_PDRj · 2024-07-12

**Soundness:** 3
**Presentation:** 3
**Contribution:** 2
**Rating:** 6
**Confidence:** 3

**Summary:**

This work proposes to incorporate the energy $U$ into the gradient-based sampling techniques. In particular, it proposes to choose the lowest energy particle as the leader and then add an extra elastic tension between the leader and followers in the Hamiltonian Monte Carlo method.

**Strengths:**

The idea is simple and clear, the toy examples are easy to understand and demonstrate the benefit of the proposed method well. In addition, the authors conduct experiments for each of the three challenging sampling scenarios identified by the authors.

**Weaknesses:**

It might worth including the overhead of the proposed method, how much slower the algorithm is per iteration compared to HMC for instance.

The tension coefficient $\lambda$ is critical, setting it to 0 recovers the baseline. But I did not find an ablation over the $\lambda$, is it hard to choose? From my understanding, if you set $\lambda$ pretty large it might recover something like gradient descent and the sampling will collapse.

**Questions:**

1. How is $\lambda$ determined per experiment? Is it possible to include an ablation over it for the three scenarios at least?

2. In Figure 7, it is unclear to me which method is better. It will be better to provide some quantitative measurement like FID or IS scores?

**Limitations:**

Yes.

---

> ### Author Rebuttal · Authors · 2024-08-07
>
> Thank you for taking the time to review our paper and for providing constructive feedback. We appreciate your comments and have addressed your points and questions below:
>
> 1. **Simplicity and Clarity:**
>    We are pleased that you found our approach simple and clear. The goal was to design a method that effectively incorporates energy information into gradient-based sampling techniques while remaining intuitive. We believe this clarity helps in demonstrating the benefits of our approach through the examples provided.
>
> 2. **Overhead and Computational Cost:**
>    Thank you for pointing out the need to include a discussion on the overhead of our method. We have made a discussion on both communitation and computation cost in the overall response.
>
> 3. **Tension Coefficient Analysis:**
>    The tension coefficient is indeed critical to our method, and we apologize for not including an ablation study on this parameter. We have now provided a table of the values we explored and a brief discussion in the overall response.
>
> 5. **Determination of the Tension Coefficient:**
>    The tension coefficient is determined empirically based on preliminary experiments. We agree that providing an ablation study for this coefficient across the three scenarios would strengthen our findings, and we are currently conducting this analysis. The results will be included in the revised manuscript.
>
> 6. **Clarity in Figure 7:**
>    Thank you for your suggestion regarding Figure 7. We acknowledge that the current presentation could be more clear and we have included quantitative metrics in the overall response.
>
> #### Limitations
> We acknowledge that the limitations section was brief. We will expand this section to discuss the scenarios where our method may face challenges, as well as ethical considerations and potential applications.
>
> ---
> Thank you once again for your valuable feedback and constructive suggestions. We are confident that the revisions and additional analyses will address your concerns and improve the paper.

---

> > ### Comment · Reviewer_PDRj · 2024-08-11
> > **Reply to Authors Response**
> >
> > I thank the authors for their clarification and additional quantitative evaluations. I look forward to the ablation studies on the tension coefficient. I will keep my original rating.

---

> ### Author Response · Authors · 2024-08-12
>
> Thank you for your feedback and for acknowledging our additional quantitative evaluations. We appreciate your interest in the ablation studies on the tension coefficient.
>
> As mentioned in the third paragraph of the "3. Sensitivity to Hyperparameters" section of the overall summary above, we have included a table displaying the hyperparameters we explored. Additionally, we have conducted an ablation study to address your concerns.
>
> ---
> ### Ablation Study
> Due to time constraints, we will briefly discuss the selection of $\lambda$, and for simplicity, we will consider updating the momentum directly by $- \eta \nabla f(x) + \lambda (x-z)$. Notice that as $\lambda \rightarrow 0$ FHL performs similarly to vanilla HMC method. As mentioned in the overall summary, determining an optimal value for $\lambda$ is challenging because it depends on the interplay of various hyperparameters and the characteristics of the sampled function. Thus, we will consider the case where $- \eta \nabla f(x) + \lambda (x-z)$ is a better choice than $- \eta \nabla f(x)$.
>
> ### Observation
> We have two key observations from both theoretical and experimental results:
> - Setting $\lambda$ to a small value may enhance performance.
> - Increasing the number of particles in the group, $n$, is likely to improve performance.
>
> **Theoretical Results**
> It is well-known that the Newton method typically converges faster than the gradient method (the Newton method exhibits quadratic convergence near the optimal solution while the gradient method has a linear convergence rate for strongly convex function). The Newton method suggests a direction given by $d_N(x) = -(\nabla^2 f(x) + \epsilon I)^{-1}\nabla f(x)$, where $\epsilon$ is a small value added to prevent the inversion of a singular matrix. Hence, we consider it desirable to have search directions that are close to $d_N$.
>
> Let $\theta(u, v)$ denote the angle between vectors $u$ and $v$, and let $\cos(u,v) = \frac{u^T v}{||u|| \cdot ||v||}$. Define the leading direction with respect to a leader $z$ as $d_{\lambda}(x;z) = -\eta \nabla f(x) + \lambda (x-z)$. Consider the positive definite quadratic function $f(x) = \frac{1}{2} x^T A x$, which results in $d_G(x) = -\eta \nabla f(x) = -\eta A x$ and $d_N(x) = -x$. For the upcoming theorems, we define a cone with axis $d$ and angle $\theta_c$ as $\text{cone}(d, \theta_c) = { x : x^T d \geq 0 \text{ and } \theta(x,d) \leq \theta_c }$. Additionally, we define $\theta_c(x) = \theta(d_N(x), d_G(x))$.
>
> **Proposition 1**: If $\lambda_1 > \lambda_2$ and $\theta(d_{\lambda_1}(x;z), d_N(x)) < \theta(d_G(x), d_N(x))$, then $\theta(d_{\lambda_2}(x;z), d_N(x)) < \theta(d_G(x), d_N(x))$. In particular, if $z \in C$ and $0 < \lambda < 1$, then $d_{\lambda}(x; z) \in C$.
>
> **Proposition 2** Define  the outward vector for a point $x$ as $N_x = d_N - \frac{<d_N,\ d_G>}{||d_G||}\cdot \frac{d_G}{||d_G||}$. If $z$ satisfies $-N_x^T (z-x) < 0$, then for sufficiently small positive $\lambda$, $d_\lambda (x; z) \in \text{cone}(d_N(x), \theta_c(x))$.
>
> **Claim 3** There exists a region that consistently improves performance. If each particle within a group of size $n$ converges to the distribution $\pi(x) = \frac{1}{Z} e^{-\frac{1}{2}x^T A x}$ (where $Z$ is a normalizing factor), then as the number of particles $n$ increases, the probability of finding an improving leader for an arbitrary particle $\tilde{x}$ increases. Specifically, this probability is given by
>
> $1 - \left(\frac{1}{Z} \int_{ z \in (z \mid f(z) < c(\tilde{x}) ) } e^{-\frac{1}{2}z^T A z} \ dz\right)^n,$
>
> where
>
> $c(\tilde{x}) =  \frac{|| N_{ \tilde{x} } ||_2^4} {2 \cdot v(\tilde{x})} ,$
>
> and
>
> $v(\tilde{x}) = N^T_{\tilde{x}} A^{-1} N_{\tilde{x}}.$
>
> **Experimental Results**
>
> We initialize $A$ as a positive-definite diagonal matrix with diagonal elements randomly selected from a uniform distribution in the range $(0.2, 20)$. The dimension of $A$ is $2000$ and we then run FHT for $2000$ iterations, adjusting $\lambda$ by dividing it by the number of particles (using $\lambda_n = \lambda / n$ instead of $\lambda$). We report the value of the leader $z$, $f(z)$, for different values of $\lambda$ and $n$.
>
> |  | $\lambda = 0.0$ (HMC) | $\lambda = 0.01$ | $\lambda = 0.02$ | $\lambda = 0.04$ | $\lambda = 0.08$ |
> | --------- | ---------------- | ---------------- | ---------------- | ---------------- | ---------------- |
> | $n = 2$ | 406.7             | **397.7**        | 398.4            | 404.8            | 411.8            |
> | $n = 4$ | 399.0             | 389.5            | **384.2**        | 384.9            | 395.5            |
> | $n =8$  | 397.7             | 391.5            | 386.7            | 380.7            | **379.6**        |
> | $n=16$  | 395.1            | 391.5            | 388.0            | 382.8            | **376.0**        |
>
> The experimental results confirm the consistency with our theoretical observations.
>
> ---
> Please let us know if you have further questions or need additional details.

---

> > ### Author Response · Authors · 2024-08-14
> >
> > - *Proof of Proposition 1.* This can be easily demonstrated by noting that $d_{\lambda_1}(x;z)$ lies within the cone $C$ formed by $\text{cone}(d_N(x), \theta_c(x))$, with $-\eta \nabla f(x)$ positioned on the boundary of $C$. Due to the convexity of $C$ and the fact that the point $d_{\lambda_2}(x;z)$ lies on the line segment connecting  $d_{\lambda_1}(x;z)$ and $-\eta \nabla f(x)$, we notice that  $d_{\lambda_2}(x;z)$ lies in the cone as well.
> >
> > - *Proof of Proposition 2.* We first define $\delta = z -x$ and $g(\lambda) = \cos(\theta(d_G + \lambda \delta, d_N)) = \frac{d_N^T (d_G+\lambda \delta)}{||d_N||\ ||d_G + \lambda \delta||}$. Then take the derivative w.r.t to $\lambda$
> > $$
> > \nabla g(\lambda) = \frac{d_N^T \delta}{||d_N||\ ||d_G + \lambda \delta||} - \frac{d_N^T(d_G+\lambda \delta)}{2 ||d_N||} [(d_G + \lambda \delta)^T (d_G + \lambda \delta)]^{-3/2} \cdot 2(d_G + \lambda \delta)^T \delta = \frac{d_N^T \delta ||d_G + \lambda \delta||^2 - d_N^T (d_G+\lambda \delta) (d_G + \lambda \delta)^T \delta}{||d_N||\ ||d_G + \lambda \delta||^3}
> > $$
> > Notice that
> > $$
> > \nabla g(\lambda)|_{\lambda = 0} = \frac{d_N^T \delta ||d_G||^2 - d_N^T d_G d_G^T \delta}{||d_N||\ ||d_G||^3} = \frac{1}{||d_N||\ ||d_G||} [d_N - \frac{<d_N,\ d_G>}{||d_G||} \cdot \frac{d_G}{||d_G||}]^T \delta > 0
> > $$
> > From Proposition 2, it can also be observed that at least half of the points in $E = \{z \mid f(z) < f(x)\}$ contribute to improving convergence (to illustrate this, consider a hyperplane defined by $H = (h \mid N_x^T h = N_x^T (z - x))$ that intersects an ellipsoid given by $(z \mid z^T A z = f(x))$). In this case, the volume of $E \cap H$ is greater than half the volume of $E$.
> >
> > - *Proof of Claim 3.* To demonstrate this, we seek to find values for $\alpha$ and $y$ such that $\alpha A y = N_x$ and $y^T N_x = ||N_x||_2^2$. This gives us $\alpha = \frac{N_x^T A^{-1} N_x}{||N_x||_2^2}$,
> >
> >    leading to the point $y = \frac{1}{\alpha} A^{-1} N_x$. The function value at $y$ is $f(y) = \frac{1}{2}y^T A y = \alpha^{-2} N_x^T A^{-1} N_x = \frac{||N_x||_2^4}{2 \cdot N_x^T A^{-1} N_x}$.
> >
> >    We can assert that for all $z \in \{z \mid f(z) < f(y)\}$, the direction of convergence could be improved by taking a step $d_{\lambda}(x;z)$ instead of $d_G(x)$ when $\lambda$ is relatively small. Consequently, for an arbitrary point $\tilde{x}$ within a group of $n$ particles, the probability of not finding any improving leader is $1 - \left(\frac{1}{Z} \int_{z \in \{z \mid f(z) < f(y)\}} e^{-\frac{1}{2}z^T A z} \, dz\right)^n$.

---

### Official Review · Reviewer_eV5W · 2024-07-17

**Soundness:** 3
**Presentation:** 4
**Contribution:** 3
**Rating:** 7
**Confidence:** 4

**Summary:**

This paper presents an interesting new approach for improving sampling methods for energy-based generative models and score-matching models. The key idea is to incorporate zeroth-order information (energy values) in addition to the typical first-order gradient information used by most sampling algorithms like Hamiltonian Monte Carlo (HMC).

The authors identify several challenging scenarios where relying solely on gradients can be problematic - cases of instability, metastability, and pseudo-stability. They argue that incorporating energy values can help mitigate issues in these situations and improve sampling efficiency and quality.

Overall, the core idea of leveraging zeroth-order information in addition to gradients is quite novel and the FHL algorithm is an elegant way to implement this for improving sampling efficiency and quality. The paper is well-motivated, the method is clearly explained, and the empirical results are compelling.

**Strengths:**

1. Novel idea of incorporating zeroth-order energy information into sampling algorithms like HMC, which typically only use gradients. This can help address issues like instability and metastability.

2. The Follow Hamiltonian Leader (FHL) algorithm is an elegant way to exchange both energy and gradient information across parallel sampling chains in a principled manner.

3. Thorough experimental evaluation across synthetic examples illustrating the identified challenging scenarios of instability, metastability, and pseudo-stability.

4. Promising results showing improved sampling quality over baselines for energy-based generative models on real datasets like CLEVR.

5. Clear motivation and well-explained methodology.

**Weaknesses:**

1. It would be better to show exploration of the sensitivity to key hyperparameters like the number of parallel sampling chains.

2. Discussion of computational cost/overhead compared to baseline sampling methods are missing in the manuscript.

**Questions:**

1. While the empirical results are compelling, can you provide any theoretical analysis or guarantees about the convergence properties, sampling quality, or stationary distribution of the FHL algorithm? Even approximate bounds or insights would strengthen the theoretical grounding.

2. What are the main computational and memory overheads introduced by running parallel chains in FHL compared to standard HMC? Is there a limit on scalability to high dimensions or is there a way to reduce overhead through approximations?

**Limitations:**

Based on the provided paper, the authors do not appear to have explicitly discussed the limitations or potential negative societal impacts of their work. The paper is primarily focused on presenting the technical details of the proposed Follow Hamiltonian Leader (FHL) sampling algorithm and its empirical evaluation.

---

> ### Author Rebuttal · Authors · 2024-08-07
>
> Thank you for your thoughtful and detailed comments on our paper. We appreciate your feedback and are grateful for the opportunity to clarify some aspects of our work. Below, we address your comments and questions:
>
>
> 1. **Novelty of Zeroth-Order Energy Information**
>    We are glad you found the incorporation of zeroth-order energy information into sampling algorithms a novel and promising idea. We believe this approach offers a fresh perspective on improving sampling stability and efficiency for sampling, particularly in challenging scenarios.
>
> 2. **Sensitivity to Key Hyperparameters**
>    We agree that exploring the sensitivity to key hyperparameters, such as the number of parallel sampling chains, is essential for a comprehensive understanding of the FHL algorithm. We have now provided a table of the values we explored and a brief discussion in the overall response.
>
> 3. **Discussion of Computational Cost/Overhead for Parallel Sampling**
>    We greatly appreciate your recognition of the elegance of the FHL algorithm across parallel sampling chains. Our aim was to develop a principled approach that effectively leverages both types of information to enhance sampling performance. We would like to point out that our method only broadcasts the leader's parameters, making it communication-efficient.
>
>    Furthermore, our method is compatible with advanced techniques like parallel tempering, as they are orthogonal approaches. We are also exploring methods to minimize overhead, such as dimensionality reduction and lazy communication.
>
> 4. **Theoretical Analysis and Quantitive Results**
>    Thank you for requesting a theoretical analysis of the convergence properties and sampling quality of the FHL algorithm. We have included convergence guarantees comparable to those offered by most sampling methods. Additionally, we provide further quantitative results for our CIFAR10 experiment (Section 5.2 in the paper).
>
> 5. **Large-scale Experiment**
>    Scalability to high-dimensional spaces is an important aspect of our algorithm. We have included a large-scale experiment in the overall response, showing that FHL can generate biomolecule conformations despite the high number of degrees of freedom and many energy landscape barriers. This sampling experiment is not only extensive but also highly significant for real-world biomolecular simulations.
>
>
> #### Limitations
>
> - **Discussion on Limitations and Societal Impacts**
>   We acknowledge that our paper did not explicitly discuss limitations or potential negative societal impacts. In the revised version, we will include a dedicated section addressing these concerns.
>
> ---
>
> We sincerely thank you for your constructive feedback, which has provided valuable insights into areas for improvement. We are committed to addressing these points in our revised manuscript and believe these enhancements will strengthen our work.

---

### Author Rebuttal · Authors · 2024-08-07

We thank all the reviewers and provide a comprehensive rebuttal to address common questions raised by several reviewers regarding the proposed FHL algorithm. Here is a brief summary:

1. Theoretical Guarantee
2. Quantitative Evaluation
3. Sensitivity to Hyperparameters
4. Parallelizability
5. Large-Scale Experiment

### 1. Theoretical Guarantee
Given the limited time available to respond to the reviewers, we have provided a sketch of the proof for the basic invariant distribution preservation property (Fixed Point Equation).

**Theorem 1:** Consider that the density $\pi$ has full support on the state space. If there is always only one unique leader in the selection steps in Algorithm 1, then Algorithm 2 preserves the invariance of the distribution $\pi$, i.e.,

$$
\pi(ds') = \int p(ds' | s) \pi(s) ds.
$$

### 2. Quantitative Evaluation
First, we want to clarify that our paper primarily focuses on sampling techniques, with generative models being just one potential application of our work. Since the FID and IS scores are mainly influenced by the quality of the generative model itself, our primary interest is in how FHL samples from a given function, even if the model is poorly trained. Therefore, we present quantitative results based on the critic provided by the model itself from our experiments on CIFAR-10.

We used the pretrained model provided in [1] and found that directly sampling by $f_\theta(x)[y]$ produces more robust results ($f_\theta$ is the neural network, $x$ is a sample, and $y$ is the class label). We report both the *value* $f_\theta(x)[y]$ and the *percentage* $\frac{\exp(f_\theta(x)[y])}{\sum_{y'} \exp(f_\theta(x)[y'])} \times 100\%$ as our metrics, where a larger value indicates better performance.

| Class    | HMC (value / percentage)  | FHL (value / percentage)  |
| -------- | ------------------------- | ------------------------- |
| Airplane |  0.724 / 99.71%           | **0.892 / 99.89%**        |
| Bird     | -0.131 / **99.99%**       | **0.194** / 99.41%        |
| Dog      |  0.094 / 95.53%           | **0.213 / 98.13%**        |
| Frog     | -0.818 / **97.33%**       | **-0.733** / 95.77%       |

Based on the visual results shown in the paper (Figure 7, page 8) and the quantitative results mentioned above, it is evident that samples from FHL achieve significantly higher $f_\theta(x)[y]$ values compared to HMC. Notably, the visual results also demonstrate that samples from FHL effectively highlight the main body of the target class object, aligning with the findings presented in this study.

> [1] Will Grathwohl, Kuan-Chieh Wang, Joern-Henrik Jacobsen, David Duvenaud, Mohammad Norouzi, and Kevin Swersky. Your classifier is secretly an energy-based model and you should treat it like one. In International Conference on Learning Representations, 2020.

### 3. Sensitivity to Hyperparameters
We experimented with different numbers of particles per group, specifically n = \{2, 4, 8, 16\}, and tested the pulling strength $\lambda$ = \{0.1, 0.01\}.

While a minimum of 2 particles per group might perform adequately, increasing the number of particles per group can enhance convergence as expected.

Our findings indicated that the pulling strength $\lambda$ behaves differently across datasets and requires fine-tuning for each one. Several factors determine the choice of pulling strength $\lambda$:
- The norm of the gradients and the spectrum of the Hessian matrix for $U(x)$.
- The temperature $T$ in the Gibbs distribution definition, $e^{-U(x) / T}$, when sampling (in our experiments, we set $T=1$, but for a more general case, $T$ could be a different value).
- The desired size of the search scope determines how far the particle may move away from the leader.

We thank the reviewers for highlighting this point, and we recognize it as an intriguing area for future research.


### 4. Parallelizability
Note that the FHL algorithm can be run in parallel. Here we discuss both the computational and communication costs in practice:

- **Computational Cost:** Although the computational cost grows linearly with the number of particles, the algorithm can be executed in parallel, allowing the total execution time to remain unchanged. Additionally, our method can be combined with techniques like multiple learning rates or different temperatures to accelerate the sampling process, as long as at least one Markov chain converges to the target distribution.

- **Communication Cost:** In modern computer architectures, NVIDIA GPUs utilize a software library called NCCL. This library is directly integrated into PyTorch and is easy to use. FHL requires only *broadcasting* the leader's parameters, which typically has **logarithmic complexity**, making it more efficient than standard collective communication methods like *all-reduce* (commonly used in Data Parallel), which have **linear complexity**.

Thus, the computational and communication costs associated with our method are efficiently managed.

### 5. Large-Scale Experiment: Study of Biomolecules Using Dialanine Peptide
*Sampling* bottlenecks have been a persistent issue in generating biomolecule conformations due to the high number of degrees of freedom and numerous energy landscape barriers. Typically, sampling becomes trapped in local minima for extended periods, making it time-consuming to explore the various conformations of a biomolecule. We implemented FHL on a real-world biomolecule, alanine dipeptide, in a vacuum at room temperature to approximate its Boltzmann distribution and used MCMC as a controlled experiment to evaluate our method's effectiveness.

We conclude that: (1) FHL is **capable of crossing energy barriers** and sampling the entire conformational space, and (2) FHL demonstrates **higher sampling efficiency** compared to MCMC.

---
We have included the theoretical and experimental results in one attached PDF. Everything is written in Markdown except for one page of figures to keep the rebuttal panel clean.

---

### Author Response · Authors · 2024-08-14

Dear Reviewers,

We sincerely appreciate the valuable contributions you have made throughout the review process. Your feedback has been instrumental in enhancing the quality of our paper, and we are committed to incorporating these improvements in future revisions.

During the rebuttal and discussion phases, we carried out additional experimental and theoretical analyses. We believe the new experiment demonstrated both the efficiency and scalability of our algorithm, while the theoretical analysis not only validated its convergence but also offered deeper insights into its remarkable performance.

- **Reviewer eV5W**: Thank you for your positive feedback. We hope that our detailed explanation of the algorithm’s complexity, as mentioned in the overall summary, addresses your concerns. Additionally, the extended analysis on hyperparameters, as discussed in our response to *Reviewer PDRj*, should further clarify our algorithm. We have also included a theoretical analysis of the algorithm's correctness, as you suggested.

- **Reviewer PDRj**: Thank you for your encouraging feedback and for acknowledging our additional quantitative evaluations. We have provided both theoretical and experimental analyses on the hyperparameters as part of our ablation study, which we hope sufficiently addresses your concerns.

- **Reviewer N2ad**: We appreciate your constructive comments and apologize for any lack of clarity in our original presentation. In response, we have expanded our explanation of the algorithm's robustness and provided a more detailed theoretical analysis of its convergence, as well as the reasons behind its superior performance (please refer to the overall summary and our response to Reviewer PDRj). Additionally, we conducted large-scale experiments to further address your concerns. We hope these improvements lead to a more favorable evaluation.

- **Reviewer M1Sx**: Thank you for increasing your score, and we are glad to have clarified your concerns. We hope that our explanation strengthens the evaluation and contribution of our work.

Furthermore, we wish to emphasize the importance of developing advanced sampling methods and their significance for real-world applications. In the biomolecule experiments we presented, variations in protein structure directly alter their functionality. To fully understand these variations, it’s essential to sample all possible protein states, especially those with lower energy (i.e., more stable states). This makes sampling a wide range of protein states critical for accurate analysis.

Traditional sampling methods often face difficulties in overcoming energy barriers, causing them to get stuck in local minima. However, our algorithm, FHL, significantly outperforms the baseline MCMC approach by **effectively crossing these barriers and exploring a much broader energy landscape**. This capability is key to better capturing the full range of possible protein states and their associated functions.

Once again, we sincerely thank all the reviewers for their comments and valuable feedback.

Best regards,
*Authors*

---

### Decision · Program_Chairs · 2024-09-25

**Decision:**

Reject

**Comment:**

The paper proposes incorporating zeroth-order information into samplers. Building on the multiple-chain Hamiltonian Monte Carlo (HMC) framework, the paper introduces a leader particle with the lowest energy to guide the convergence of all particles. While the reviewers found the proposed algorithm interesting and promising, they raised concerns about the lack of theoretical analysis and the need for more experimental results.

- The main idea of Elastic Leapfrog is to include an additional L2 difference term. This concept has already been explored in the sampling literature (see, e.g., [1, 2, 3]). The paper should revise its language to avoid overclaiming novelty and include a discussion of these related works.

[1] Structured logconcave sampling with a restricted gaussian oracle. COLT 2021

[2] Entropy-MCMC: Sampling from Flat Basins with Ease. ICLR 2024

[3] Diffusive Gibbs Sampling. ICML 2024

- The baselines in the paper currently include only standard Langevin dynamics and Hamiltonian Monte Carlo. Given the numerous recent advancements in improving sampler's convergence, the paper should incorporate a broader range of modern samplers as baselines. Additionally, the proposed method is closely related to replica exchange, and the paper should also include replica exchange and its recent variants as baselines for comparison.

- The paper lacks theoretical analysis, which is crucial for a sampling algorithm. Demonstrating both asymptotic and non-asymptotic convergence is important. Furthermore, it would be beneficial to compare the convergence bounds of the proposed method with those of existing methods.

- The algorithm's design is not thoroughly convincing. The new method is introduced in Section 4, but the description is brief and lacks the theoretical or empirical evidence needed to support the design.

The authors provided a new theorem without detailed proofs and additional empirical results during the rebuttal. I encourage the authors to continue improving the paper, particularly in the areas of related work, theoretical analysis, and a more comprehensive empirical comparison.